# UNCONSTRAINED ROBUST ONLINE CONVEX OPTIMIZATION

## ABSTRACT

This paper addresses online learning with "corrupted" feedback. Our learner is provided with potentially corrupted gradients $\tilde{g}_t$ instead of the "true" gradients $g_t$. We make no assumptions about how the corruptions arise: they could be the result of outliers, mislabeled data, or even malicious interference. We focus on the difficult "unconstrained" setting in which our algorithm must maintain low regret with respect to any comparison point $\|u\| \in \mathbb{R}^d$. The unconstrained setting is significantly more challenging as existing algorithms suffer extremely high regret even with very tiny amounts of corruption (which is not true in the case of a bounded domain). Our algorithms guarantee regret $\|u\|G(\sqrt{T} + k)$ when $G \geq \max_t \|g_t\|$ is known, where $k$ is a measure of the total amount of corruption. When $G$ is unknown we incur an extra additive penalty of $(\|u\|^2 + G^2)k$.

## 1 INTRODUCTION

In this paper, we consider unconstrained online convex optimization (OCO) under the presence of adversarial corruptions. In general, OCO is a framework in which a learner iteratively outputs a prediction $w_t \in \mathcal{W}$, then observes a vector $g_t = \nabla \ell_t(w_t)$ for some convex loss function $\ell_t : \mathcal{W} \to \mathbb{R}$, and then incurs a loss of $\ell_t(w_t)$. The learner's performance over a time horizon $T$ is evaluated by the *regret* relative to a fixed competitor $u \in \mathcal{W}$, denoted as $R_T(u)$

$$R_T(u) := \sum_{t=1}^{T} \langle g_t, w_t - u \rangle \geq \sum_{t=1}^{T} \ell_t(w_t) - \ell_t(u)$$

The inequality above follows by convexity of $\ell_t$. Classical results in this field consider a bounded domain $\mathcal{W}$ with known diameter $D$ and a Lipschitz bound $G \geq \max_t \|g_t\|$. In this setting, the standard minimax-optimal result is $R_T(u) \leq O(GD\sqrt{T})$ (Zinkevich, 2003; Abernethy et al., 2008).

Our work focuses on the *unconstrained* case $\mathcal{W} = \mathbb{R}^d$, where it is typical to aim for a regret guarantee that scales not with a uniform diameter bound $D$, but with the norm of the comparator $\|u\|$. Such bounds are often called "comparator adaptive" (because they adapt to the comparator $u$), or "parameter-free" (because this adaptivity suggests that the algorithms require less hyperparameter tuning). In this unconstrained setting, the classical algorithms achieve $R_T(u) = \tilde{O}(\|u\|G\sqrt{T})$ (Mcmahan & Streeter, 2012; McMahan & Orabona, 2014; Orabona & Pál, 2016; Orabona, 2014) (which is also optimal).

We are interested in a harder variant of the OCO framework with "corrupted" gradients. Specifically, instead of any direct information about the function $\ell_t$, after each round the learner is provided with a vector $\tilde{g}_t$ that should be interpreted as an estimate of $g_t = \nabla \ell_t(w_t)$. Our aim is to obtain a regret that scales as $\|u\|G(\sqrt{T} + k)$ for all $u \in \mathcal{W}$, where $k$ is some measure of the degree to which $\tilde{g}_t \neq g_t$ that will be formally defined in Section 2. Roughly speaking, $k$ can be interpreted as the number of rounds in which $\tilde{g}_t \neq g_t$. Notably, the desired rate is robust to adversarial corruptions in the sense that it allows $k = O(\sqrt{T})$ before the bound becomes worse than the optimal result *without* corruptions.

Our dual challenges of corrupted $\tilde{g}_t$ and unconstrained $\mathcal{W}$ are naturally motivated by problems in practice. The unconstrained setting is ubiquitous in machine learning - consider the classical logistic regression setting, for which it is unusual to impose constraints. The corrupted $\tilde{g}_t$ in contrast is less commonly studied, but represents a common practical issue: the computed gradients may not be good

estimates of a "true" gradient, either due to the presence of statistical outliers, numerical precision issues in the gradient computation, or mislabeled or otherwise damaged data.

We distinguish two different settings in our results: one in which the algorithm is provided with prior knowledge of a number $G \geq \max_t \|g_t\|$, and one in which it is not. This is a common dichotomy in unconstrained OCO, even without corruptions. In the former case, the classical result of $\tilde{O}(\|u\|G\sqrt{T})$ is obtainable, while in the latter case it is not: instead the optimal results are $R_T(u) \leq \tilde{O}(\|u\| \max_t \|g_t\| \sqrt{T} + \|u\|^3 \max_t \|g_t\|)$ (Cutkosky, 2019a; Mhammedi & Koolen, 2020), or $\tilde{O}(\|u\| \max_t \|g_t\| \sqrt{T} + \|u\|^2 + \max_t \|g_t\|^2)$ by Cutkosky & Mhammedi (2024). The later excels particularly whenever $G$ is not excessively large: $G \leq \|u\|\sqrt{T}$.

To the best of our knowledge, the setting of unconstrained OCO with corruptions has not been studied before. Perhaps the closest works to ours are Zhang & Cutkosky (2022); Jun & Orabona (2019); van der Hoeven (2019) and van Erven et al. (2021). Zhang & Cutkosky (2022); Jun & Orabona (2019); van der Hoeven (2019) study the unconstrained setting, but assume that $\tilde{g}_t$ is a random value with $\mathbb{E}[\tilde{g}_t] = g_t$. In contrast, we assume no such stochastic structure on $\tilde{g}_t$. On the other hand, van Erven et al. (2021) does not make any assumptions about the nature of the corruptions, but assumes tha $\mathcal{W}$ has finite diameter $D$. They considers an outlier corruption model: $\bar{S} = \{t \in [T] : g_t \neq \tilde{g}_t\}$ and its complement $S = [T] \setminus \bar{S}$. Thus $S$ represents rounds with outliers occurred. The online learner receives $\tilde{g}_t$ with only the knowledge of $|\bar{S}| \leq k$, algorithm developed achieves $R_S(u) := \sum_{t \in S} \langle g_t, w_t - u \rangle \leq O(DG(\sqrt{T} + k))$ by skipping evaluations on outlier rounds. Our development will borrow some ideas from van Erven et al. (2021) with the aim to bound $R_T(u)$ without skipping evaluations, but it turns out that the unconstrained domain provides unique challenges that we must overcome, as detailed in Section 3.

The notion of adversarial corruption is common in the field of robust statistics, with early efforts focusing primarily on the presence of outliers in linear regression (Huber, 2004; Cook, 2000; Thode, 2002). These inspired broader application in machine learning, asuch as Robust PCA (Candès et al., 2011), anomaly detection (Raginsky et al., 2012; Delibalta et al., 2016; Zhou & Paffenroth, 2017; Sankararaman et al., 2022), robust regression (Klivans et al., 2018; Cherapanamjeri et al., 2020; Chen et al., 2022), and mean estimation (Lugosi & Mendelson, 2021). For a comprehensive review of recent advances in this area, see Diakonikolas & Kane (2019).

Adversarial corruption also significantly impacts iterative algorithms other than OCO, prompting considerable theoretical research within the framework of stochastic bandits (Lykouris et al., 2018; Gupta et al., 2019; Ito, 2021; Agarwal et al., 2021) and stochastic optimization (Chang et al., 2022; Sankararaman & Narayanaswamy, 2024).

**Contributions and Organization**  In the case that the algorithm is given prior knowledge of $G$, we provide an algorithm that achieves $R_T(u) = \tilde{O}(\|u\|G(\sqrt{T} + k))$ in Section 4.1, with a matching lower bound (see Section 4.2). Alternatively, when $G$ is unknown, a regret bound with an additional penalty of $(\|u\|^2 + G^2)k$ is attained (see Section 5.2).

Meanwhile, we provide two specific applications of our results in Sections 4.3. First, we show that our method can be used to solve stochastic convex optimization problems in some of the gradient computations are altered in an arbitrary way. Second, we solve a natural "online" version of a distributionally robust optimization problem. Before providing our main results, we introduce notation and define our corruption model in Section 2.

## 2 NOTATION AND PROBLEM SETUP

**Notation**  We consider $\ell_t : \mathcal{W} \to \mathbb{R}$ as a convex function, where we consider $\mathcal{W} = \mathbb{R}^d$. Let $w_t \in \mathcal{W}$ be iterates from some online learning algorithm and denote $g_t = \nabla \ell_t(w_t)$ as the "true" (sub)gradient. Let $\tilde{g}_t$ be the the possibly corrupted that is observable to the learner. Define $\mathbb{1}\{\cdot\}$ as the indicator function, where $\mathbb{1}\{\text{TRUE}\} = 0, \mathbb{1}\{\text{FALSE}\} = 0$. Use $|\cdot|$ to denote the cardinality of a set, which counts the number of elements in the set, and occasionally we use it as the absolute value of real numbers. Let $\|\cdot\|$ denote the Euclidean norm. Denote $\mathbb{R}^+ = \{x \in \mathbb{R} : x \geq 0\}$. We define shorthand notation for sets $[T] = \{1, 2, \ldots, T\}$ and $[a, T] = \{a, a + 1, \ldots, T\}$ for some $a \in [T]$. We use

$\mathcal{B} \subseteq [T]$ to denote an index set, and $\bar{\mathcal{B}} = [T] \setminus \mathcal{B}$ for its complement. We use $O(\cdot)$ to hide constant factors and $\tilde{O}(\cdot)$ to additionally conceal any polylogarithmic factors.

**Problem Setup** Instead of the true gradients $g_t$, we our algorithms only receive potentially corrupted gradients $\tilde{g}_t$. Two natural measures to quantify corruptions are:

$$k_{\text{count}} := \sum_{t=1}^{T} \mathbb{1}\{g_t \neq \tilde{g}_t\} \quad (1) \qquad k_{\text{deviation}} := \frac{1}{G} \sum_{t=1}^{T} \|g_t - \tilde{g}_t\| \quad (2)$$

where $G$ is a scalar that satisfies $G \geq \max_t \|g_t\|$ and is often referred to as the "Lipschitz constant". The metric $k_{\text{count}}$ counts the rounds in which $\tilde{g}_t \neq g_t$ but allowing for arbitrarily large deviations $\|\tilde{g}_t - g_t\|$ in those rounds. This is suitable for detecting outlier effects and highlighted in studies such as van Erven et al. (2021); Sankararaman & Narayanaswamy (2024). Conversely, $k_{\text{deviation}}$ measures the cumulative deviation, accommodating corruption in every round, making it optimal for identifying subtle yet widespread errors or malicious activities, akin to the issues addressed in Lykouris et al. (2018); Gupta et al. (2019); Ito (2021); Agarwal et al. (2021); Chang et al. (2022).

In order to provide a unified way to study those two distinct corruption measures in Equation (1) and (2), we assume that our algorithm is provided with a number $k$ that satisfies:

$$|\mathcal{B}| := |\{t \in [T] : \|g_t - \tilde{g}_t\| \geq G\}| \leq k \quad (3) \qquad \frac{1}{G} \sum_{t=1}^{T} \min(\|g_t - \tilde{g}_t\|, G) \leq k \quad (4)$$

where $\mathcal{B}$ particular denotes rounds of corruption with a big deviation. Notice that

$$|\mathcal{B}| \leq \min(k_{\text{count}}, k_{\text{deviation}}) \qquad \frac{1}{G} \sum_{t=1}^{T} \min(\|g_t - \tilde{g}_t\|, G) \leq \min(k_{\text{count}}, k_{\text{deviation}})$$

Hence, it suffices to design algorithms remain robust with a given $k$ satisfies Equation (3) and (4) where $k$ can be set either as $k_{\text{count}}$ or $k_{\text{deviation}}$ for appropriate type of corruptions that is encountering.

## 3 CHALLENGES IN UNCONSTRAINED DOMAIN

Dealing with corruptions with an unconstrained domain is significantly more challenging than one with a bounded domain - even if the corruptions are so "small" that $\|g_t - \tilde{g}_t\| \leq G$. In a bounded $\mathcal{W}$ with a diameter $D$, an algorithm that completely ignores the possibility of corruptions and directly runs on $\tilde{g}_t$ may have low regret. This can be seen as follows: since $\|u - w_t\| \leq D$ for every $u, w_t \in \mathcal{W}$, we have:

$$\sum_{t=1}^{T} \langle g_t, w_t - u \rangle \leq \sum_{t=1}^{T} \langle \tilde{g}_t, w_t - u \rangle + \sum_{t=1}^{T} \|g_t - \tilde{g}_t\| \|w_t - u\| \leq \sum_{t=1}^{T} \langle \tilde{g}_t, w_t - u \rangle + kGD$$

In this case, $\|u - w_t\| \leq D$ prevents the algorithm from straying too far from the comparator $u$.

The situation is much more difficult in the *unconstrained* setting. Algorithm for this setting typically produce outputs $w_t$ that potentially grow *exponentially fast* in order to quickly compete with comparators that are very far from the starting point. However, this also means the algorithm is especially fragile to corruption since the growth of $w_t$ can be highly sensitive to deviations in $\|g_t - \tilde{g}_t\|$. Even a small deviation could cause $w_t$ to move extremely far away and therefore incur a very high regret. This phenomenon is illustrated in Figure 1 with the KT-bettor algorithm Orabona & Pál (2016), which is a standard example of an unconstrained learner.

In Figure 1, we considered $\ell_t(w) = |x - 1|$ for all $t$. Figure 1a and 1b demonstrate $k = 20$ gradients being corrupted by setting $\tilde{g}_t = -g_t$ during rounds $t \in [300, 300 + k - 1] = [300, 319]$ over a time span of $T = k^2 = 400$. This results in an exponential deviation away from the comparator $u = 1$ and so incurs a high regret. Finally, we show that this problem becomes exacerbated as $k$ increases by simulating $k \in [20, 30, 40, 50, 60, 70]$ for $T = k^2$ in figure 1c.

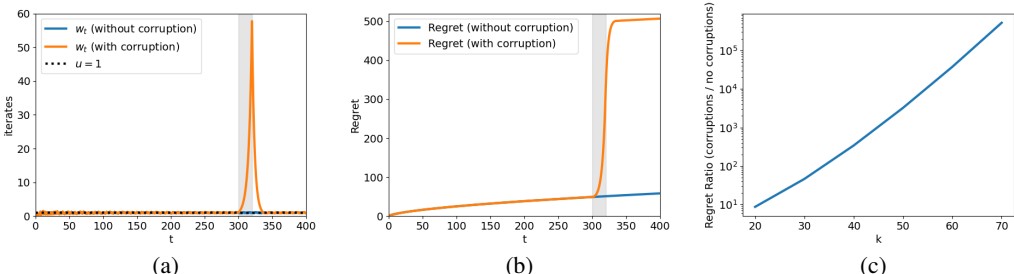

Figure 1: KT-bettor with $\ell(w) = |w - 1|$ and comparator $u = 1$ (a)-(b): $T = 400$ and corruption happens during $t \in [300, 319]$. (c): Ratio between Regret with corruptions and without corruptions with various total corrupted rounds $k \in [20, 30, 40, 50, 60, 70]$ and $T = k^2$.

In receiving possibly corrupted gradients $\tilde{g}_t$, our general approach is to first employ a gradient clipping step with some threshold $h_t$ that outputs a "clipped" version $\tilde{g}_t^c$, defined as follows:

$$\tilde{g}_t^c = \frac{\tilde{g}_t}{\|\tilde{g}_t\|} \min\left(h_t, \|\tilde{g}_t\|\right) \tag{5}$$

This preprocessing step "corrects" some corruption effect when $h_t$ is appropriately chosen. For example, in the case of $h_t = G \geq \max_t \|g_t\|$, then $\tilde{g}_t^c$ is always "less corrupted" than $\tilde{g}_t$, as $\|\tilde{g}_t^c - g_t\| \leq \|\tilde{g}_t - g_t\|$. Then $\tilde{g}_t^c$ is used as a feedback to an online learner, yielding the following expression for $R_T(u)$:

$$R_T(u) := \sum_{t=1}^{T} \langle g_t, w_t - u \rangle = \sum_{t=1}^{T} \langle \tilde{g}_t^c, w_t - u \rangle + \sum_{t=1}^{T} \langle g_t - \tilde{g}_t^c, w_t - u \rangle \tag{6}$$

After this preprocessing, we design an algorithm that controls both of the above summations, even without ever seeing the true gradients $g_t$. Depends on whether $G \geq \max_t \|g_t\|$ is known or not, the treatment to both steps differs. We introduce our developments for known and unknown $G$ in Section 4.1 and 5, respectively. Although our analysis only focused on $\mathcal{W} = \mathbb{R}$ in those sections, a dimension-free black box reduction from Cutkosky & Orabona (2018) facilitates the adaptation of our approach to $\mathcal{W} = \mathbb{R}^d$ as discussed in Appendix D.

## 4 ROBUST LEARNING WITH KNOWLEDGE OF LIPSCHITZ CONSTANT

In this section, we proceed under the assumption that $G \geq \max_t \|g_t\|$ is known a priori. We therefore will set $h_t = G$ for all iterations in the definition $\tilde{g}_t^c$ (see Equation 5).

### 4.1 THE ALGORITHM AND REGRET GUARANTEE

As motivated in Section 3, Equation (5) is a preprocessing step on $\tilde{g}_t$ with $h_t = G$, thus outputs $\tilde{g}_c$ as a feedback to online learner. The regret in Equation (6) can be further upper bounded as:

$$R_T(u) \leq \sum_{t=1}^{T} \langle \tilde{g}_t^c, w_t - u \rangle + \left(\max_t |w_t| + |u|\right) \left(\sum_{t \in \mathcal{B}} |g_t - \tilde{g}_t^c| + \sum_{t \in \bar{\mathcal{B}}} |g_t - \tilde{g}_t^c|\right)$$

$$\leq \sum_{t=1}^{T} \langle \tilde{g}_t^c, w_t - u \rangle + \left(\max_t |w_t| + |u|\right) \left(\sum_{t \in \mathcal{B}} |g_t - \tilde{g}_t| + G|\bar{\mathcal{B}}|\right)$$

$$\leq \sum_{t=1}^{T} \langle \tilde{g}_t^c, w_t - u \rangle + kG \left(\max_t |w_t| + |u|\right) \tag{7}$$

where $\mathcal{B}$ is defined in Equation (3), $\bar{\mathcal{B}} = [T] \setminus \mathcal{B}$. The second line is due to $|g_t - \tilde{g}_t^c| \leq |g_t - \tilde{g}_t| \leq G, \forall t \in \bar{\mathcal{B}}$. The last inequality is due to the corruption model presented in Equation (4).

The main challenge arises from the term $kG \max_t |w_t|$ in Equation (7), which could be extremely large (potentially exponential in $t$ as shown in Lemma 8 Zhang & Cutkosky (2022)). Even if $\max_t |w_t|$

is bounded by $O(\sqrt{T})$, a worst-case scenario with $k = O(\sqrt{T})$ could still yield linear regret. This issue is reminiscent of challenges identified by Zhang & Cutkosky (2022), who studied *stochastic* corruptions with $\mathbb{E}[\tilde{g}_t] = g_t$. Taking inspiration from their solution, we consider a composite loss function $\tilde{\ell}_t(w) = \langle \tilde{g}_t^c, w \rangle + r_t(w)$, where $r_t : \mathcal{W} \to \mathbb{R}^+$ is convex. By feeding $\nabla \tilde{\ell}_t(w_t)$ to an online learner, the following relation reveals through rearrangement and the convexity of $\tilde{\ell}_t$:

$$\sum_{t=1}^{T} \langle \tilde{g}_t^c, w_t - u \rangle = \sum_{t=1}^{T} \tilde{\ell}_t(w_t) - \tilde{\ell}_t(u) - r_t(w_t) + r_t(u) \leq \sum_{t=1}^{T} \langle \nabla \tilde{\ell}_t(w_t), w_t - u \rangle - r_t(w_t) + r_t(u)$$

Thus the true regret $R_T(u)$ can be decomposed as:

$$R_T(u) \leq \underbrace{\sum_{t=1}^{T} \langle \tilde{g}_t^c + \nabla r_t(w_t), w_t - u \rangle}_{\text{goal 1: } R_T^{\mathcal{A}}(u) \leq \tilde{O}(|u|G\sqrt{T})} + \underbrace{kG \max_t |w_t| - \sum_{t=1}^{T} r_t(w_t)}_{\text{goal 2: OFFSET} \leq \tilde{O}(1)} + \underbrace{kG|u| + \sum_{t=1}^{T} r_t(u)}_{\text{goal 3: MAINTAIN} \leq \tilde{O}(|u|Gk)} \quad (8)$$

Equation (8) suggests that if we could choose $\mathcal{A}$ and $r_t$ such that $R_T^{\mathcal{A}}(u) \leq \tilde{O}(|u|G\sqrt{T})$, OFFSET is $O(1)$ and MAINTAIN is $\tilde{O}(|u|Gk)$, this would imply $R_T(u) \leq \tilde{O}(|u|G(\sqrt{T} + k))$. We choose $r_t$ from a family of Huber losses first proposed by Zhang & Cutkosky (2022) and displayed in Equation (9) with $c = kG, \alpha = \epsilon/kG$:

$$r_t(w; c, \alpha) = \begin{cases} c(\ln T|w| - (\ln T - 1)|w_t|) \frac{|w_t|^{\ln T - 1}}{(\sum_{i=1}^{t} |w_i|^{\ln T} + \alpha^{\ln T})^{1 - 1/\ln T}}, & |w| > |w_t| \\ c|w|^{\ln T} \frac{1}{(\sum_{i=1}^{t} |w_i|^{\ln T} + \alpha^{\ln T})^{1 - 1/\ln T}}, & |w| \leq |w_t| \end{cases} \quad (9)$$

This $r_t(w)$ has two important properties: polynomial growth when $|w| \leq |w_t|$ and linear growth otherwise. The polynomial growth ensures $\sum_t r_t(w_t)$ is large enough to ensure OFFSET $= O(1)$. The linear growth is slow enough to prevent $\sum_t r_t(u)$ from blowing, ensuring MAINTAIN $\leq \tilde{O}(|u|Gk)$. Both bounds are provided in Lemma 8, Appendix C.

With the specified $r_t$, the final step is to design an algorithm $\mathcal{A}$ that ensures $R_T^{\mathcal{A}}(u) \leq \tilde{O}(|u|G\sqrt{T})$. On the surface, this may seem straightforward as $R_T^{\mathcal{A}}$ involves the *observed* value $\tilde{g}_t^c + \nabla r_t(w_t)$ rather than the *unobserved* values $g_t$. One might therefore hope to simply apply a standard OCO algorithm out-of-the-box. Unfortunately, $\tilde{g}_t^c + \nabla r_t(w_t)$ may be as a large as $G + k$, and so such an approach would yield only $R_T^{\mathcal{A}}(u) \leq O(|u|(G + k)\sqrt{T})$. Fortunately, we known how the choice of $w_t$ will influence $\nabla r_t(w_t)$. This suggests applying tools from optimistic online learning (Rakhlin & Sridharan, 2013), whose regret depends only on the "unpredictable" component of the loss sequence (i.e. $\tilde{g}_t^c$). We employ the optimism framework of Cutkosky (2019b). This requires two algorithms, $\mathcal{A}_1, \mathcal{A}_2$, which must both be online learners obtaining the optimal rate in parameter-free literature (e.g.: Mhammedi & Koolen (2020); Jacobsen & Cutkosky (2022); Zhang et al. (2024)). At a high level, $\mathcal{A}_1$ is run "as normal", while $\mathcal{A}_2$ is responsible for "correcting" the output of $\mathcal{A}_1$ to exploit with the known form of $\nabla r_t(w_t)$. See Appendix B for details. Note that standard optimistic methods require $\nabla r_t(w_t)$ to not depend on $w_t$ and so do not immediately apply; we employ a modification inspired by Zhang & Cutkosky (2022) to account for this.

Our algorithm and analysis for $\mathcal{W} = \mathbb{R}$ is specified in Algorithm 1 and Theorem 1. The straightforward extension to $\mathcal{W} = \mathbb{R}^d$ is provided in Theorem 11, which essentially replaces the $|u|$ in Theorem 1 with $\|u\|$ for $\mathcal{W} = \mathbb{R}^d$ with no dependence on $d$.

---

**Algorithm 1** Robust Online Learning in Unconstrained Domain with $G$

---

**Require:** Time horizon $T$, Lipschitz consntant $G$. Two independent online learnering algorithms $\mathcal{A}_1, \mathcal{A}_2$ with optimal rate in parameter-free literature (e.g.: Mhammedi & Koolen (2020)) where a concrete example is the assumption in Theorem 7 (they can be the same algorithm). Corruption parameter $k$. Base algorithm parameters $\epsilon$. Regularization relevant parameters: $c, \alpha$.
1: **Initialize:**
    Initialize $\mathcal{A}_1, \mathcal{A}_2$ with $\epsilon$.
2: **for** $t = 1$ to $T$ **do**
3:     Receive $x_t$ from $\mathcal{A}_1$
4:     *# The next steps "correct" $x_t$ via our modified optimistic update.*

5:     Recieve $y_t$ from $\mathcal{A}_2$
6:     Solve for $w_t$: $w_t = x_t - y_t \nabla r_t(w_t)$
7:     *# End optimism correction.*
8:     Play $w_t$, suffer loss $\langle g_t, w_t \rangle$, receive $\tilde{g}_t$
9:     Compute $\tilde{g}_t^c$ through Equation (5) with $h_t = G$.
10:    Compute regularizer $r_t(w; c, \alpha)$ as defined in Equation (9)
11:    Send $\tilde{g}_t^c + \nabla r_t(w_t)$, and $(1 + k \ln T)G$ to $\mathcal{A}_1$
12:    Send $-\langle \tilde{g}_t^c + \nabla r_t(w_t), \nabla r_t(w_t) \rangle$, and $(1 + k \ln T)^2 G^2$ to $\mathcal{A}_2$
13: **end for**

**Theorem 1.** *Suppose $g_t, \tilde{g}_t$ satisfies assumptions in Equation (3) and (4). Set $c = kG, \alpha = \frac{\epsilon}{kG}$ for some $\epsilon > 0$. For $T \geq 3$, Algorithm 1 runs on $\tilde{g}_t^c$ guarantees*

$$R_T(u) \leq \tilde{O}\left[\epsilon + |u|G\left(\sqrt{T} + k\right)\right]$$

Theorem 1 shows that the penalty for corrupted gradients is at most $\tilde{O}(|u|Gk)$. This result has a few intriguing properties. First, so long as $k \leq \sqrt{T}$, the penalty is subasymptotic to the standard uncorrupted regret bound $\tilde{O}(|u|G\sqrt{T})$. That is, we can tolerate $k$ up to $\sqrt{T}$ essentially "for free". Next, observe that for $u = 0$, the regret is $\epsilon$ no matter what $k$ is. Constant regret at the origin is typical for unconstrained algorithms, but is especially remarkable for our corrupted setting. Imagine a scenario in which we define 0 to represent some "default" action. Our bound then suggests that *no matter how much corruption is present*, we never do significantly worse than this default.

### 4.2 LOWER BOUNDS

We present a lower bound in Theorem 2 with proofs deferred in Appendix E. This result shows that the upper bound of Theorem 1 is tight. In addition, we provide a second lower bound as Theorem 16 in Appendix E, which has the matching log factor.

**Theorem 2.** *For every $D > 0$, there exists a comparator $u^* \in \mathbb{R}^d$ such that $\|u^*\| = D$, $\tilde{g}_1, \cdots, \tilde{g}_T$ and $g_1, \cdots, g_T$ such that $\|g_t\|, \|\tilde{g}_t\| \leq 1$, $\sum_{t=1}^{T} \mathbb{1}\{\tilde{g}_t \neq g_t\} = k$:*

$$\sum_{t=1}^{T} \langle g_t, w_t - u^* \rangle \geq \Omega\left[\|u^*\|\left(\sqrt{T} + k\right)\right]$$

### 4.3 EXAMPLES

Here, we provide implication of Algorithm 1 to stochastic convex optimization and distributionally robust optimization. Example illustrated also applies to $\mathcal{W} = \mathbb{R}^d$.

**Stochastic convex optimization with corruptions**   OCO and convex stochastic optimization are connected through the classical Online-to-Batch Conversion Orabona (2019). Below, we present the implications of Theorem 1 stochastic convex optimization in a setting where $k$ gradient evaluations are arbitrarily corrupted.

**Corollary 3** (Stochastic Convex Optimization via Online to Batch)**.** *Suppose $\mathcal{L} : \mathcal{W} \to \mathbb{R}$ is convex and $\mathbb{E}[\ell_t(w)] = \mathcal{L}(w), g_t = \nabla \ell_t(w_t)$ and $\mathbb{E}_t[g_t] \leq G$. Algorithm 1 have access to $\tilde{g}_t$ such that $\sum_{t=1}^{T} \mathbb{1}\{g_t \neq \tilde{g}_t\} \leq k$, then Algorithm 1 guarantees*

$$\mathbb{E}\left[\mathcal{L}\left(\frac{\sum_{t=1}^{T} w_t}{T}\right) - \mathcal{L}(u)\right] \leq \tilde{O}\left[\frac{\epsilon + |u|G\left(\sqrt{T} + k\right)}{T}\right]$$

*Proof.* The proof leverages the standard online to batch conversion (Theorem 3.1 in Orabona (2019) by setting $\alpha_t = 1$), then combining with the regret bounds from Theorem 1.   □

**Distributionally robust optimization** Distributionally robust optimization is a form of robust stochastic optimization on training data sampled from distribution $P$ that is not the same as the population distribution $Q$ (Ben-Tal et al., 2009; 2015). Typically, $Q$ is considered as uniform, but the actual training data collection process might be biased, meaning $P$ is different to $Q$. In this situation, stochastic optimization which treats each training example with equal weight is no longer appropriate.

Namkoong & Duchi (2016) formalized this framework as the following model with respect to a set of losses $\ell_1, \ldots \ell_T$, and an uncertainty set $\mathcal{P}_k = \{P \in \Delta^T : D_f(P||Q) \leq C(k,T)\}$, where $D_f(P||Q)$ is the $f$-Divergence, for a convex function $f : \mathbb{R}^+ \mapsto \mathbb{R}$ with $f(1) = 0$.

$$\arg\min_w \sup_{P \in \mathcal{P}_k} \sum_{t=1}^T p_t \ell_t(w)$$

the decision variable from above formulation takes account into the worst case distributional uncertainty, hence is intuitively associated with improving generalization error given an appropriate uncertainty set $\mathcal{P}_k$ (Sagawa et al., 2019).

Distributionally robust optimization is increasingly relevant in the training of large language models, where training data are sourced from different domains (Xie et al., 2023). This is due to data from some domain are relatively atypical in comparison to others in representing the overall population distribution (Oren et al., 2019). Although empirical gain has been observed by incorporating distributionally robust optimization, the scalability has always been a primary concern for model training (Levy et al., 2020; Qi et al., 2021). Therefore, we consider a natural "online" version of distributionally robust optimization model proposed by Namkoong & Duchi (2016), with its online analogous metric formulated as:

$$\sup_{P \in \mathcal{P}_k} \sum_{t=1}^T p_t(\ell_t(w_t) - \ell_t(u))$$

We present the implication of Algorithm 1 to this problem with respect to total variation $D_{TV}$ and Kullback-Leibler divergence $D_{KL}$. In particular, we assume $\ell_t$ is convex and $Q$ is uniform.

**Corollary 4** (Online Distributionally Robust Optimization). *Suppose $\tilde{g}_t \in \nabla \ell_t(w_t)$ and $|\tilde{g}_t| \leq G$. Algorithm 1 runs on $\tilde{g}_t$ guarantees*

$$\sup_{P \in \mathcal{P}_k} \sum_{t=1}^T p_t(\ell_t(w_t) - \ell_t(u)) \leq \tilde{O}\left[\frac{\epsilon + |u|G\left(\sqrt{T} + k\right)}{T}\right]$$

*for $D_{TV} \leq \frac{k}{T}$. In addition, in the case where $D_{KL} \leq \frac{2k^2}{T^2}$ the same guarantee is achieved.*

*Proof.* We begin with the case of $D_{TV}(P||Q) = \frac{1}{2}\sum_{t=1}^T q_t |\frac{p_t}{q_t} - 1| \leq \frac{k}{T}$, where $q_t = \frac{1}{T}$. First, we link the regret incurred by Algorithm 1 that runs on $g_t$, and we denote the *unobservable* gradient as $\tilde{g}_t = \frac{p_t}{q_t} g_t$

$$\sum_{t=1}^T p_t(\ell_t(w_t) - \ell(u)) \leq \sum_{t=1}^T p_t \langle g_t, w_t - u \rangle = \sum_{t=1}^T q_t \langle g_t, w_t - u \rangle + \sum_{t=1}^T q_t \left(\frac{p_t}{q_t} - 1\right) \langle g_t, w_t - u \rangle$$

$$= \frac{1}{T}\left(\sum_{t=1}^T \langle g_t, w_t - u \rangle + \sum_{t=1}^T \langle \tilde{g}_t - g_t, w_t - u \rangle\right)$$

since $\frac{1}{G}\sum_{t=1}^T \|g_t - \tilde{g}_t\| \leq \sum_{t=1}^T |1 - \frac{p_t}{q_t}| \leq 2k$, $\tilde{g}_t, g_t$ satisfies Equation (2), hence Theorem 1 provides the guarantee:

$$\sum_{t=1}^T p_t(\ell_t(w_t) - \ell(u)) \leq \tilde{O}\left[\frac{\epsilon + |u|G\left(\sqrt{T} + k\right)}{T}\right]$$

In terms of $D_{KL}$, we exploit the Pinsker's inequality $D_{TV} \leq \sqrt{2D_{KL}}$, Hence $D_{KL} \leq \frac{2k^2}{T^2}$ yields to the same results. $\square$

## 5 ROBUST LEARNING WITH UNKNOWN LIPSCHITZ CONSTANT

In this section, we consider $G \geq \max_t \|g_t\|$ is unknown. Since we do not know $G$, we cannot set $h_t = G$ for all $t$ as in Section 4. So, we first develop an alternative approach to learn $h_t$ on-the-fly in order to supply Equation (5) as a pre-processing step. Then we show an compatible algorithm in maintaining small true regret $R_T(u)$ as defined in Equation (6).

### 5.1 ADAPTIVE THRESHOLDING

In this section, we introduce the two "tracking mechanisms" FILTER (Algorithm 6) and TRACKER (Algorithm 7) and the parameters $\alpha_t, \beta_t$ as defined in Equation (11) and (12). These mechanisms and quantities form the foundation for algorithm design to achieve desired regret bound in Section 5.2.

The corruption model in Equation (3) naturally restricts the number of "big" $\tilde{g}_t$, since it implies that at most $k$ values of $t$ can have $\|\tilde{g}_t\| > 2G$ (See Lemma 17). Based on this observation, we draw inspiration from van Erven et al. (2021) and propose a simple way to learn a "threshold" $h_t$ on-the-fly which provides an estimate of $G$. This mechanism is named as FILTER and is displayed as Algorithm 6 in Appendix F.

FILTER maintains a "checkpoint" $h$ which serves as a rough estimate of the future clipping threshold $h_{t+1}$. Both the threshold $h_t$ and check point $h$ start with some initial value $\tau_G > 0$. The checkpoint $h$ remains the same until $k + 1$ instances where $\|\tilde{g}_t\| \geq h$ are observed, at which point $h$ is doubled. At iterations in which a single $\|\tilde{g}_t\| \geq h$ is observed, the threshold is finely adjusted as $h_{t+1} = h_t + h/(k+1)$. The thresholds $h_1, \cdots, h_T$ are supplied to (5) to truncate $\tilde{g}_t$ to $\tilde{g}_t^c$ such that $\|\tilde{g}_t^c\| \leq h_t$.

Notice that $h$ only doubles if it is guaranteed that some $g_t$ satisfies $h \leq \|g_t\|$, so that at most $O(k \log_2 G/\tau_G)$ rounds have $h \leq \|g_t\|$. Denote rounds where gradients are clipped as $\bar{\mathcal{P}} = \{t \in [T] : \tilde{g}_t \neq \tilde{g}_t^c\}$, the doubling criterion in $h$ allows FILTER to guarantee $|\bar{\mathcal{P}}| \leq \tilde{O}(k)$ (See Lemma 18). This means only a small fraction of $\tilde{g}_t$ are truncated when $h_t$ has not yet became a good lower bound estimate in $G$. This FILTER strategy improves upon a method with a similar purpose in van Erven et al. (2021); it uses only constant space rather than $O(k)$ space.

Using FILTER, we can decompose the regret in Equation (6) by using $\tilde{g}_t = \tilde{g}_t^c$ for $t \in \mathcal{P}$:

$$R_T(u) = \sum_{t=1}^{T} \langle \tilde{g}_t^c, w_t - u \rangle + \underbrace{\sum_{t \in \mathcal{P}} |g_t - \tilde{g}_t^c| (|w_t| + |u|)}_{\text{corruption error}} + \underbrace{\sum_{t \in \bar{\mathcal{P}}} |g_t - \tilde{g}_t^c| (|w_t| + |u|)}_{\text{truncation error}} \quad (10)$$

In addition to the expected "corruption error", the price to pay for not knowing $G$ is to pick up an additional "truncation error". Thus for all $t \in \bar{\mathcal{P}}$, the learner needs to be informed that its decision $w_t$ should be decreased to guarantee the overall "truncation error" is under control. To this end, we use $h_{t+1}$ from FILTER to compute a triggering signal $\alpha_t \in [0, \gamma_\alpha]$ for a to-be-specified $\gamma_\alpha$ as shown in equation (11). This $\alpha_t$ quantity (which first appeared appeared in Cutkosky & Mhammedi (2024)) is used to specify a new regularization term that causes $w_t$ to decrease. Since $h_{t+1} > h_t$ only when $t \in \bar{\mathcal{P}}$, we have $\alpha_t > 0$ and an active regularization only at those rounds. Overall, the FILTER outputs $h_{t+1}$ in such a way as to allow $\sum_t \alpha_t = \tilde{O}(1)$ which is crucial for later algorithm design.

Taking a similar approach in managing the "truncation error", we also employed a doubling strategy to keep a rarely-changing estimate of $\max_t |w_t|$ as $z_t$, which we call TRACKER as shown in Algorithm 7, Appendix G. $\beta_t \in [0, \gamma_\beta]$ is then computed with $z_{t+1}$ as shown in Equation (12) with the property of $\beta_t > 0$ only when $w_t$ has noticeably big magnitude. Thus the interpretation of $\beta_t$ is an "alert" to an online learner that the $w_t$ value may need to decrease to prevent "corruption error" from accumulating. Similarly, the TRACKER outputs $z_{t+1}$ which allows for $\sum_t \beta_t = \tilde{O}(1)$.

$$\alpha_t = \gamma_\alpha \cdot \frac{(h_{t+1} - h_t)/h_{t+1}}{1 + \sum_{i=1}^{t}(h_{i+1} - h_i)/h_{i+1}} \quad (11) \qquad \beta_t = \gamma_\beta \cdot \frac{(z_{t+1} - z_t)/z_{t+1}}{1 + \sum_{i=1}^{t}(z_{i+1} - z_i)/z_{i+1}} \quad (12)$$

## 5.2 THE ALGORITHM AND REGRET ANALYSIS

In this section, we design an online learner $\mathcal{A}$ operating on $\mathcal{W} = \mathbb{R}$ and relying on feedback $\tilde{g}_t^c, h_{t+1}$ from FILTER, such that $|\tilde{g}_{t+1}^c| \leq h_{t+1}$. We will eventually achieve $R_T(u) \leq \tilde{O}(|u|(\sqrt{T} + k) + (|u|^2 + G^2)k)$ by integrating ingredients from the preceding sections. We begin with a simplification of Equation (10) that combinates the "corruption error" and "truncation error":

$$R_T(u) \leq \sum_{t=1}^{T} \langle \tilde{g}_t^c, w_t - u \rangle + \underbrace{(kG + |\bar{\mathcal{P}}|(G + h_T))}_{A \leq \tilde{O}(kG)} \left( \max_{t \in \mathcal{P}} |w_t| + |u| \right) \tag{13}$$

Equation (13) reveals the same problematic dependence on $\tilde{O}(kG \max_t |w_t|)$ encountered in Section 4.1. This shared challenge motivated us to take similar approach: use a regularization function $\phi_t : \mathcal{W} \to \mathbb{R}^+$ to "cancel" excess terms. The chosen $\phi_t$ is a combination of $r_t(w)$ is the same form as Equation (8) and a quadric regularizer with and $a_t = \alpha_t + \beta_t$ which were independently defined as Equation (11) and (12), respectively.

$$\phi_t(w) = r_t(w) + a_t w^2$$

This yields a regret decomposition directly through adding and subtracting in Equation (13):

$$R_T(u) \leq \underbrace{\sum_{t=1}^{T} \langle \tilde{g}_t^c, w_t - u \rangle + \sum_{t=1}^{T} \phi_t(w_t) - \phi_t(u)}_{\text{goal 5}: R_T^{\mathcal{A}}(u) \text{ small}} + \underbrace{A \max_{t \in [T]} |w_t| - \sum_{t=1}^{T} \phi_t(w_t)}_{\text{goal 4}: \text{OFFSET small}} + \underbrace{A|u| + \sum_{t=1}^{T} \phi_t(u)}_{\text{goal 6}: \text{MAINTAIN}}$$

$$\tag{14}$$

The chosen regularization $\phi_t$ allows us to achieve simultaneously: (1) MAINTAIN $\leq \tilde{O}(|u|k + |u|^2)$ and (2) OFFSET $\leq O(G^2 k)$. The former (1) is due to $\alpha_t, \beta_t = 0$ on most rounds because of the structure of FILTER and TRACKER, hence $\sum_t a_t = \sum_t \alpha_t + \sum_t \beta_t = \tilde{O}(1)$. In addition $\sum_t r_t(u)$ grows sublinearly with respect to $T$ as discussed in Section 4.1. For the latter (2), in Appendix I, we show:

$$\text{OFFSET} \lesssim A^2 \sum_{t: \alpha_t > 0} \frac{1}{\alpha_t} + A^2 \sum_{t: \beta_t > 0} \frac{1}{\beta_t}$$

Intuitively, both FILTER and TRACKER identify rounds requiring control of "truncation error" and "corruption error", and $\alpha_t > 0$ and $\beta_t > 0$ for those rounds only. The design of FILTER and TRACKER then makes the number of such rounds small.

It remains to choose a learner $\mathcal{A}$ such that $R_T^{\mathcal{A}}(u) \leq \tilde{O}(|u|G(\sqrt{T} + k) + |u|^2 k)$. Unfortunately, this $\phi_t$ is not Lipschitz, which makes applying standard tools for constructing unconstrained online learners difficult. We combat this by employing the "epigraph-based regularization" technique recently developed by Cutkosky & Mhammedi (2024) in combination with our optimistic online learning method (further explanations see Appendix H). Briefly, for any pair $(w_t, y_t)$ with $y_t \geq w_t^2$, we have:

$$R_T^{\mathcal{A}}(u) \leq \underbrace{\sum_{t=1}^{T} \langle \tilde{g}_t^c + \nabla r_t(w_t), w_t - u \rangle}_{R_T^{\mathcal{A}_w}(u)} + \underbrace{\sum_{t=1}^{T} a_t(y_t - u^2)}_{R_T^{\mathcal{A}_y}(u)}$$

This is a sum of two regrets for the pair $w_t$ and $y_t$ with Lipschitz linear losses, subject to $y_t \geq w_t^2$. We solve this problem using a pair of unconstrained learners $(\mathcal{A}_w, \mathcal{A}_y)$ that produce $(\hat{w}_t, \hat{y}_t) \in \mathbb{R}^2$ and guarantee regret $R_T^{\mathcal{A}_w}(u), R_T^{\mathcal{A}_y}(u)$. Then, we employ a black-box conversion from unconstrained-to-constrained learning due to Cutkosky & Orabona (2018) to enforce the constraint: this involves a projection $\Pi_W : \mathbb{R}^2 \to W := \{(w, y) : y \geq w^2\}$ and a certain technical correction to the gradient feedback as highlighted in Green. Finally, selecting $\mathcal{A}_w$ using a similar optimistic algorithm as in Section 4 (highlighted in Pink) and $\mathcal{A}_y$ as a standard unconstrained OCO algorithm with optimal rate allowed us to achieve the desired overall regret.

Our algorithm is specified in Algorithm 2, followed by its regret guarantee in Theorem 5 (proved in Appendix I). The extension of Algorithm 2 to $\mathcal{W} = \mathbb{R}^d$ is provided in Theorem 25.

---

**Algorithm 2** Regularization by Epigraph and Optimism

---

**Require:** Time horizon $T$, FILTER as Algorithm 6, TRACKER as Algorithm 7. An algorithm $\mathcal{A}_y$ with optimal rate in parameter-free literature (e.g.: Mhammedi & Koolen (2020)). Corruption parameter $k$. Base algorithm parameters $\epsilon$. Regularization relevant parameters: $c, \alpha$ (used to define $r_t(w)$ via Equation (9) in Line 10) and $\gamma, \gamma_\alpha, \gamma_\beta$ (used in Lines 8, 9 to define $a_t$)

1: **Initialize:**
    Initialize Algorithm 3 as $\mathcal{A}_w$ with $\epsilon$. Initialize $\mathcal{A}_y$ with $\epsilon$
    Initialize FILTER with $\tau_G$ (outputs $h_t$ as a conservative lower-bound guess for $G$ )
    Initialize TRACKER with $\tau_D$ (outputs $z_t$ as a conservative lower-bound guess for $\max_t |w_t|$).
2: **for** $t = 1$ to $T$ **do**
3:     Receive $\hat{w}_t$ from $\mathcal{A}_w$; Receive $\hat{y}_t$ from $\mathcal{A}_y$
4:     Compute Operators in Definition 20 with $h_t \leftarrow h_t + c \ln T, \gamma \leftarrow \gamma$
5:     *# Explicit projection of $(\hat{w}_t, \hat{y}_t)$ through projection map $\Pi_W^t$ as in Definition 20*
6:     Compute Projection $(w_t, y_t) = \Pi_W^t((\hat{w}_t, \hat{y}_t))$
7:     Play $w_t$, receive $\tilde{g}_t^c, h_{t+1}$ from FILTER; Send $w_t$ to TRACKER and receive $z_{t+1}$
8:     Compute $\alpha_t, \beta_t$ as defined in Equations (11, 12)
9:     Compute quadratic regularizer weights $a_t = \alpha_t + \beta_t$
10:    *# Get regularizer $r_t$ as defined Equation (9)*
11:    Compute gradient for optimism: $\nabla r_t(w_t)$
12:    *# Compute gradient correction direction $(\delta_t^w, \delta_t^y)$ with $\|\cdot\|_{*,t}$ and $\nabla S_t$ as in Definition 20*
13:    $(\delta_t^w, \delta_t^y) = \|(\tilde{g}_t^c + \nabla r_t(w_t), a_t)\|_{*,t} \nabla S_t((\hat{w}_t, \hat{y}_t))$ *# used to correct for projection (line 6)*
14:    *# Send corrected gradients / gradient for optimism*:
15:    Send $(\frac{1}{2}\tilde{g}_t^c, \frac{1}{2}h_{t+1})$ and $(\frac{1}{2}(\nabla r_t(\hat{w}_t) + \delta_t^w), \frac{3}{2}(h_{t+1} + c\ln T))$ to $\mathcal{A}_w$ *# optimism learner*
16:    Send $\frac{1}{2}(a_t + \delta_t^y)$, and $\frac{3}{2}\gamma$ to $\mathcal{A}_y$
17: **end for**

---

**Theorem 5.** *Suppose $g_t, \tilde{g}_t$ satisfies assumptions in Equation (3) and (4). Algorithm 2 in response to $\tilde{g}_t$ with parameters: $\alpha = \epsilon/c, \gamma_\alpha = \gamma_\beta = \frac{\gamma}{2}$, for some $\epsilon, c, \gamma, \tau_G, \tau_D > 0$. Then Algorithm 2 guarantees a regret bound $R_T(u)$:*

$$R_T(u) \leq \tilde{O}\left[\epsilon + |u|c + |u|\max(\tau_G, G)\sqrt{T} + |u|^2\gamma\right] + \frac{4k^2G^2}{\gamma}\ln\frac{8k^2G^2}{c\gamma\tau_D} + c\tau_D + kG\tau_D$$

$$+ \frac{4(k+1)^2(G+h_T)^2}{\gamma}\left(1 + \ln\frac{h_{T+1}}{\tau_G}\right)\max\left(\left\lceil\log_2\frac{8G}{\tau_G}\right\rceil, 1\right)$$

**Corollary 6.** *With $c = 2k/\tau_D, \gamma = k + 1$ and rest of parameters same as Theorem 5, Algorithm 2 guarantees a regret bound $R_T(u)$:*

$$\tilde{O}\left[\epsilon + k\left(1 + \frac{|u|}{\tau_D} + G\tau_D\right) + |u|\max(\tau_G, G)\left(\sqrt{T} + k\right) + \left(|u|^2 + \max\left(\tau_G^2, G^2\right)\right)(k+1)\right]$$

Just as in the known-$G$ case, the parameter settings in Corollary 6 yield $\tilde{O}(\sqrt{T})$ regret so long as $k \leq \sqrt{T}$ so that we can experience a significant amount of corruption without damaging the asymptotics of the regret bound. We can also achieve the desirable "safety" property of Theorem 1 in which the regret with respect to the baseline point $u = 0$ is constant no matter what $k$ is via a different setting of the regularization parameters provided in Corollary 24 in the appendix. However, in this case we now pay a larger penalty for $u \neq 0$ that scales with $k^2$ rather than $k$.

## 6 CONCLUSION

In this paper, we considered unconstrained online convex optimization that only have access to potentially corrupted gradients $\tilde{g}_t$ instead of the true gradient $g_t$, in which the corruption level is measured by $k$. In the case that $G \geq \max_t \|g_t\|$ is known, we provide an algorithm that achieves the optimal regret guarantee $\|u\|G(\sqrt{T} + k)$. When $G$ is unknown it incur an extra additive penalty of $(\|u\|^2 + G^2)k$. While the $\|u\|^2 + G^2$ is optimal without corruption (Cutkosky & Mhammedi, 2024), it is unclear whether the multiplicative dependence on $k$ is optimal in the presence of corruption.

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

## A  UNCONSTRAINED ONLINE CONVEX OPTIMIZATION WITH HINTS

In the unconstrained setting, there are algorithms requires a uniform bound $G \geq \max_t \|g_t\|$ upfront which guarantees $\tilde{O}(\|u\|G\sqrt{T})$ McMahan & Orabona (2014); Orabona & Pál (2016); Cutkosky & Orabona (2018); Zhang et al. (2022). In the case where $G$ is unknown, algorithms are usually devised through an intermediate step with a slightly ideal scenario, that is the algorithm receives a gradient $g_t$ with a "hints" $h_{t+1} = \max_{i \leq t+1} \|g_i\|$ at each iteration $t$. It turns out by having access to $h_t$ to guide the algorithm, same regret $\tilde{O}(\|u\|h_T\sqrt{T})$ can be achieved Cutkosky (2019a); Mhammedi & Koolen (2020); Jacobsen & Cutkosky (2022); Zhang et al. (2024).

In this paper, we also follows the same strategy of assuming a good hints $h_t = \max_{i \leq t} \|g_t\|$ is supplied to the algorithm, and eventually investigate the scenario of only the current best estimate $h_t \approx \max_{i \leq t-1} \|g_t\|$ is available. Hence most of the proofs in the appendix are displayed in the way of relying on a time varying "hints": $0 < h_1 \leq \cdots h_T \leq h_{T+1}$ to accommodate the design of both known $G$ and unknown $G$ case.

## B  OPTIMISTIC ONLINE CONVEX OPTIMIZATION

This section follows *optimistic* reduction in unconstrained setting from Cutkosky (2019b) in learning from a composite loss $\ell_t(w) = \langle g_t + r_t, w \rangle$ where $|g_t| \leq G_t$ is adversarially generated and $|r_t| \leq H_t$ is predictable or even chosen by the user. By a straightforward application of the standard unconstrained OCO algorithm out-of-the-box in responding to $g_t + r_t$, $R_T(u)$ depends on the gradient norm $\tilde{O}\left((\max_t G_t + H_t)\sqrt{T}\right)$. However, given the *optimistic* nature of $r_t$ being predictable, one should hope for algorithm should not suffer $H_T$ growing with respect to $T$. Cutkosky (2019b) achieved the desired dependence $\tilde{O}(\max_t G_t\sqrt{T} + \max_t H_t)$ by lunching two algorithms $\mathcal{A}_1$ learns $x_t$ and $\mathcal{A}_2$ learning $y_t$ and produces iterates $w_t$ as:

$$w_t = x_t - y_t r_t$$

This update is similar to the structure of online subgradient descent, where $\mathcal{A}_1$ learns an pseudo iterate $x_t \sim w_t$, and $\mathcal{A}_2$ learns a step size $y_t$ to make finer adjustment to $x_t$ by $r_t$. In the following Theorem, we make no effort in improving the result, but follow the same analysis strategy as Cutkosky (2019b) with the adaptation of base learners $\mathcal{A}_1, \mathcal{A}_2$ must be unconstrained and *Lipschitz adaptive*: that is receives $g_t, h_{t+1}$ such that $|g_t| \leq h_t$ while maintain low regret, the optimal rate (Mhammedi & Koolen, 2020; Jacobsen & Cutkosky, 2022; Zhang et al., 2024) is usually same as the assumption in Theorem 7. This procedure is fomalized as Algorithm 3:

---
**Algorithm 3** Optimistic Online Learning in Unconstrained Domain with $h_t$
---
**Require:** Time horizon $T$, Sequence $0 < G_1 \leq G_2 \leq \cdots \leq G_{T+1}$ such that $|g_t| < G_t$; $0 < H_1 \leq H_2 \leq \cdots \leq H_{T+1}$ such that $|r_t| \leq H_t$, Two independent online learning algorithms $\mathcal{A}_1, \mathcal{A}_2$ with optimal rate in parameter-free literature (e.g.: Mhammedi & Koolen (2020)) where a concrete example is the assumption in Theorem 7 (they can be the same algorithm).
1: **Initialize:**
      Initialize $\mathcal{A}_1, \mathcal{A}_2$ with $\epsilon$.
2: **for** $t = 1$ to $T$ **do**
3:      Receive $x_t$ from $\mathcal{A}_1$
4:      Receive $y_t$ from $\mathcal{A}_2$
5:      Compute: $w_t = x_t - y_t r_t$
6:      Play $w_t$, receive $(g_t, G_{t+1})$ and $(r_t, H_{t+1})$
7:      Send $g_t + r_t$, and $G_{t+1} + H_{T+1}$ to $\mathcal{A}_1$
8:      Send $-\langle g_t + r_t, r_t \rangle$, and $(G_{t+1} + H_{t+1})^2$ to $\mathcal{A}_2$
9: **end for**

---

**Theorem 7.** *Suppose $\mathcal{A}$ produces $w_t$ in response to $g_t$ such that $|g_t| \leq G_t$ and $0 \leq G_1 \leq \cdots \leq G_T$ ensures the following guarantee for a given $\epsilon > 0$:*

$$R_T^{\mathcal{A}}(u, G_T) = \sum_{t=1}^T \langle g_t, w_t - u \rangle \leq \epsilon + A|u|\sqrt{\sum_{t=1}^T |g_t|^2 \ln\left(1 + \frac{|u|T^C G_T}{\epsilon G_1}\right)} + BG_T|u|\ln\left(1 + \frac{|u|T^C G_T}{\epsilon G_1}\right)$$

*for all $u \in \mathcal{W}$ and for some positive constants $A, B, C$. Initiate two independent copy $\mathcal{A}$ denote as $\mathcal{A}_1$ and $\mathcal{A}_2$, and suppose there is another sequence $r_t$ such that $|r_t| \leq H_t$ and $0 \leq H_1 \leq \cdots \leq H_T$. $\mathcal{A}_1$ produces $x_t$ in response to $g_t + r_t$ and $\mathcal{A}_1$ produces $y_t$ in response to $-\langle g_t + r_t, r_t \rangle$. Then with $w_t = x_t - y_t r_t$. Then*

$$\sum_{t=1}^{T} \langle g_t + r_t, w_t - u \rangle \leq \tilde{O}\left[ \epsilon + |u|\sqrt{\sum_{t=1}^{T} g_t^2} + |u|(G_T + H_T) \right]$$

*Proof.*

$$\sum_{t=1}^{T} \langle g_t + r_t, w_t - u \rangle = \sum_{t=1}^{T} \langle g_t + r_t, x_t - u \rangle + y_t(-\langle g_t + r_t, r_t \rangle)$$

$$= R_T^1(u) + R_T^2(y_*) - y_* \sum_{t=1}^{T} \langle g_t + r_t, r_t \rangle$$

$$\leq \inf_{y_* \geq 0} R_T^1(u) + R_T^2(y_*) - y_* \sum_{t=1}^{T} \langle g_t + r_t, r_t \rangle$$

First, we substitute $R_T^1(u), R_T^2(y_*)$. For $R_T^1(u)$, since $\mathcal{A}_1$ runs on $g_t + r_t$ and $|g_t + r_t| \leq G_T + H_T$ for all $t$. Hence we should set $g_t \leftarrow g_t + r_t$, $G_T \leftarrow G_T + H_T$. Similarly for $R_T^2(u)$, where we run $\mathcal{A}_2$ on $-\langle g_t + r_t, r_t \rangle$ and is uniformly bounded by $(G_T + H_T)H_T$.

$$\leq \inf_{y_* \geq 0} 2\epsilon + A|u|\sqrt{\sum_{t=1}^{T} |g_t + r_t|^2 \ln\left(1 + \frac{|u|T^C(G_T + H_T)}{\epsilon(G_1 + H_1)}\right)}$$

$$+ B(G_T + H_T)|u|\ln\left(1 + \frac{|u|T^C(G_T + H_T)}{\epsilon(G_1 + H_1)}\right)$$

$$+ Ay_*\sqrt{\sum_{t=1}^{T} \langle g_t + r_t, r_t \rangle^2 \ln\left(1 + \frac{y_* T^C(G_T + H_T)H_T}{\epsilon(G_1 + H_1)H_1}\right)}$$

$$+ B(G_T + H_T)H_T y_* \ln\left(1 + \frac{y_* T^C(G_T + H_T)H_T}{\epsilon(G_1 + H_1)H_1}\right)$$

$$- y_* \sum_{t=1}^{T} \langle g_t + r_t, r_t \rangle$$

For the last term, use $-2\langle a, b \rangle = |a - b|^2 - |a|^2 - |b|^2$

$$= \inf_{y_* \geq 0} 2\epsilon + A|u|\sqrt{\sum_{t=1}^{T} |g_t + r_t|^2 \ln\left(1 + \frac{|u|T^C(G_T + H_T)}{\epsilon(G_1 + H_1)}\right)}$$

$$+ B(G_T + H_T)|u|\ln\left(1 + \frac{|u|T^C(G_T + H_T)}{\epsilon(G_1 + H_1)}\right)$$

$$+ Ay_*\sqrt{\sum_{t=1}^{T} \langle g_t + r_t, r_t \rangle^2 \ln\left(1 + \frac{y_* T^C(G_T + H_T)H_T}{\epsilon(G_1 + H_1)H_1}\right)}$$

$$+ B(G_T + H_T)H_T y_* \ln\left(1 + \frac{y_* T^C(G_T + H_T)H_T}{\epsilon(G_1 + H_1)H_1}\right)$$

$$+ \frac{y_*}{2} \sum_{t=1}^{T} |g_t|^2 - |g_t + r_t|^2 - |r_t|^2$$

To get a clearer view of the expression, we denote

$$\Delta_1 = \ln\left(1 + \frac{|u|T^C(G_T + H_T)}{\epsilon(G_1 + H_1)}\right), \qquad \Delta_2 = \ln\left(1 + \frac{|u|T^C(G_T + H_T)H_T}{\epsilon(G_1 + H_1)H_1}\right)$$

and $X = \sum_{t=1}^T |g_t + r_t|^2$. Further, $\sum_{t=1}^T \langle g_t + r_t, r_t \rangle^2 \leq \sum_{t=1}^T |g_t + r_t|^2 |r_t|^2 \leq H_T^2 X$ where the last step we applied a uniform bound $|r_t| \leq H_T$. Thus

$$\sum_{t=1}^T \langle g_t + r_t, w_t - u \rangle \leq 2\epsilon + A|u|\sqrt{X\Delta_1} + B|u|(G_T + H_T)\Delta_1$$

$$+ \inf_{y_* \geq 0} Ay_* H_T\sqrt{X\Delta_2} + By_*(G_T + H_T)H_T\Delta_2 + \frac{y_*}{2}\sum_{t=1}^T |g_t|^2 - |r_t|^2 - \frac{y_*}{2}X$$

$$(15)$$

It remains to select the correct $y^* \in \mathbb{R}^+$ so the expression on the right hand side of Equation (15) balances to the desired result. It turns out the optimal selection of $y^*$ depends on whether $\Delta_1 < \Delta_2$ is true or not. Fortunately, $y_*$ will eventually vanish from the right hand side, so we can select $y_*$ by cases and then combing the results.

1. First, we assume $\Delta_2 < \Delta_1$. In this case, select

$$y_* = \min\left(\frac{|u|}{H_T}, \frac{2A|u|\sqrt{\Delta_1}}{\sqrt{\max\left(0, \sum_{t=1}^T |g_t|^2 - |r_t|^2\right)}}\right)$$

   Substitute the choice of $y^*$ to Equation (15). In particular, explicitly using the first argument for the third to last summand, and take the second argument to balance the second to the last term. We keep $y^*$ at other places for convenience.

$$\sum_{t=1}^T \langle g_t + r_t, w_t - u \rangle \leq 2\epsilon + A|u|\sqrt{X\Delta_1} + B|u|(G_T + H_T)\Delta_1$$

$$+ Ay_* H_T\sqrt{X\Delta_2} + B|u|(G_T + H_T)\Delta_2$$

$$+ A|u|\sqrt{\Delta_1 \max\left(0, \sum_{t=1}^T |g_t|^2 - |r_t|^2\right)} - \frac{y_*}{2}X$$

$$\leq 2\epsilon + B|u|(G_T + H_T)(\Delta_1 + \Delta_2) + A|u|\sqrt{\Delta_1 \sum_{t=1}^T |g_t|^2}$$

$$+ \sup_{X \geq 0} A|u|\sqrt{X\Delta_1} - \frac{y_*}{4}X + \sup_{Z \geq 0} Ay_* H_T\sqrt{Z\Delta_2} - \frac{y_*}{4}Z$$

   For the last two terms $x \mapsto K\sqrt{x} + \frac{y_*}{4}x$ for $K, y_* > 0$ attains its maximum at $\sqrt{x} = \frac{2K}{y_*}$ yields to $\frac{K^2}{y_*}$

$$\leq 2\epsilon + B|u|(G_T + H_T)(\Delta_1 + \Delta_2) + A|u|\sqrt{\Delta_1 \sum_{t=1}^T |g_t|^2}$$

$$+ \frac{A^2|u|^2\Delta_1}{y^*} + A^2 y_* H_T^2 \Delta_2$$

   It remains to determine the correct upper bound with the selected $y_*$. For the term contains $y^*$ at the denominator, use $\frac{x}{\min(a,b)} \leq \frac{x}{a} + \frac{x}{b}$ for $x, a, b \geq 0$. For the remaining term involves $y_*$, use $y_* \leq \frac{|u|}{H_T}$. Hence

$$\sum_{t=1}^T \langle g_t + r_t, w_t - u \rangle \leq 2\epsilon + B|u|(G_T + H_T)(\Delta_1 + \Delta_2) + A|u|\sqrt{\Delta_1 \sum_{t=1}^T |g_t|^2}$$

$$+ A^2|u|H_T\Delta_1 + \frac{A}{2}|u|\sqrt{\Delta_1 \max\left(0, \sum_{t=1}^T |g_t|^2 - |r_t|^2\right)} + A^2|u|H_T\Delta_2$$

$$\leq 2\epsilon + B|u|(G_T + H_T)(\Delta_1 + \Delta_2)$$

$$+ \frac{3A}{2}|u|\sqrt{\Delta_1 \sum_{t=1}^T |g_t|^2} + A^2|u|H_T(\Delta_1 + \Delta_2) \tag{16}$$

2. Now consider $\Delta_2 \geq \Delta_1$ and set

$$y_* = \min\left(\frac{|u|}{H_T}, \frac{2A|u|\sqrt{\Delta_2}}{\sqrt{\max\left(0, \sum_{t=1}^T |g_t|^2 - |r_t|^2\right)}}\right)$$

and follows the identical algebra as the first case, we have

$$\sum_{t=1}^T \langle g_t + r_t, w_t - u\rangle \leq 2\epsilon + B|u|(G_T + H_T)(\Delta_1 + \Delta_2)$$

$$+ \frac{3A}{2}|u|\sqrt{\Delta_2 \sum_{t=1}^T |g_t|^2} + A^2|u|H_T(\Delta_1 + \Delta_2) \tag{17}$$

Combining both cases of Equation (16) and ( 17), we have :

$$\sum_{t=1}^T \langle g_t + r_t, w_t - u\rangle \leq 2\epsilon + B|u|(G_T + H_T)(\Delta_1 + \Delta_2)$$

$$+ \frac{3A}{2}|u|\sqrt{\max(\Delta_1, \Delta_2)\sum_{t=1}^T |g_t|^2} + A^2|u|H_T(\Delta_1 + \Delta_2)$$

substitute $\Delta_1, \Delta_2$ and use $\tilde{O}(\cdot)$ to hide log factors then we have the desired result. $\qquad\square$

## C  BOUNDS ON REGULARIZER AND THEOREM 1

The development of Algorithm 1 and Theorem 1 was based on appropriate choice of regularizer $r_t$ which was firstly studied by Zhang & Cutkosky (2022). We include Lemma 8 by gathering relevant bounds from Zhang & Cutkosky (2022) for completeness followed by the proof of Theorem 1.

**Lemma 8** (Lemma 11 and Lemma 13 of Zhang & Cutkosky (2022)). *Let $r_t : \mathcal{W} \to \mathbb{R}^+$ be defined as follows for some $c \geq 0, \alpha > 0$ and $p \geq 1$,*

$$r_t(w; c, p, \alpha) = \begin{cases} c(p|w| - (p-1)|w_t|)\frac{|w_t|^{p-1}}{(\sum_{i=1}^t |w_i|^p + \alpha^p)^{1-1/p}}, & |w| > |w_t| \\ c|w|^p \frac{1}{(\sum_{i=1}^t |w_i|^p + \alpha^p)^{1-1/p}}, & |w| \leq |w_t| \end{cases}$$

*Then*

$$\sum_{t=1}^T r_t(w_t) \geq c\left(\left(\sum_{t=1}^T |w_t|^p + \alpha^p\right)^{1/p} - \alpha\right)$$

$$\sum_{t=1}^T r_t(u) \leq cp|u|T^{1/p}\left[\ln\left(1 + \left(\frac{|u|}{\alpha}\right)^p\right)^{(p-1)/p} + 1\right]$$

*In particular, when $p = \ln T$ for $T \geq 3$:*

$$\sum_{t=1}^T r_t(w_t) \geq c\left(\max_t |w_t| - \alpha\right)$$

$$\sum_{t=1}^{T} r_t(u) \leq 3 \ln T c |u| \left[ \ln \left( 1 + \left( \frac{|u|}{\alpha} \right)^p \right) + 2 \right]$$

*Proof.* The first set of bounds are the same as Zhang & Cutkosky (2022) Lemma 13. For the second set of bounds: the lower bound is due to $\left( \sum_{t=1}^{T} |w_t|^p + \alpha^p \right)^{1/p} \geq \left( \sum_{t=1}^{T} |w_t|^p \right)^{1/p}$ followed by an application of of Lemma 11 in Zhang & Cutkosky (2022); the upper bound is due to $x^q \leq x + 1$ for $x > 0$ and $0 < q < 1$, where we set $x = \ln \left( 1 + \left( \frac{|u|}{\alpha} \right)^p \right)$ and $q = (p-1)/p$ folowed by $T^{1/\ln T} = e \leq 3$. $\square$

**Theorem 1.** *Suppose $g_t, \tilde{g}_t$ satisfies assumptions in Equation (3) and (4). Set $c = kG, \alpha = \frac{\epsilon}{kG}$ for some $\epsilon > 0$. For $T \geq 3$, Algorithm 1 runs on $\tilde{g}_t^c$ guarantees*

$$R_T(u) \leq \tilde{O} \left[ \epsilon + |u|G \left( \sqrt{T} + k \right) \right]$$

*Proof.* The proof begins with the regret decomposition in Equation (8) and is displayed below for convenience. We aim to show each component satisfy the desired bound as follows:

$$\tilde{R}_T(u) \leq \underbrace{\sum_{t=1}^{T} \langle \tilde{g}_t^c + \nabla r_t(w_t), w_t - u \rangle}_{\text{goal1: } R_T^{\mathcal{A}}(u) \leq \tilde{O}(|u|G\sqrt{T})} + \underbrace{kG \max_t |w_t| - \sum_{t=1}^{T} r_t(w_t)}_{\text{goal2: OFFSET} \leq \tilde{O}(1)} + \underbrace{kG|u| + \sum_{t=1}^{T} r_t(u)}_{\text{goal3: MAINTAIN} \leq \tilde{O}(|u|Gk)}$$

goal1: since $|\tilde{g}_t^c| \leq h_t = G, |\nabla r_t(w_t)| \leq c \ln T = 2kG \ln T$. Thus $R_T^{\mathcal{A}}(u)$ is guaranteed by Theorem 7 by setting $G_t = h_t = G, H_t = 2k \ln T$, yields to

$$R_T^{\mathcal{A}}(u) \leq \tilde{O} \left[ \epsilon + |u| \sqrt{\sum_{t=1}^{T} |\tilde{g}_t^c|^2} + |u|(h_T + kG) \right] = \tilde{O} \left[ \epsilon + |u|G \left( \sqrt{T} + k \right) \right]$$

goal2 & goal3: both are guaranteed by Lemma 8. Specifically by substitute $c, \alpha$:

$$\text{OFFSET} \leq kG \max_t |w_t| - kG(\max_t |w_t| - \epsilon/kG) = \epsilon$$

$$\text{MAINTAIN} \leq kG|u| + 3kG \ln T |u| \left[ \ln \left( 1 + \left( \frac{|u|kG}{\epsilon} \right)^{\ln T} \right) + 2 \right] = \tilde{O}(kG|u|)$$

$\square$

## D   DIMENSION-FREE ROBUST LEARNING WITH KNOWN $G$

In this section, we aim to extend Algorithm 1 operates on $\mathbb{R}$ to $\mathbb{R}^d$ through a dimension-free reduction introduced by Cutkosky & Orabona (2018) followed by its regret guarantee. Since there are mixture of scalar and vectors, to maintain clarity we use $a$ to denote scalar and $\mathbf{a}$ to denote vector in this section.

Cutkosky & Orabona (2018) proposed the task of learning $\mathbf{w}_t \in \mathbb{R}^d$ can be distributed into two algorithms: $\mathcal{A}_{\mathbb{R}}$ to produce $x_t \in \mathbb{R}$ and $\mathcal{A}_{\mathcal{B}_d}$ to produce $\mathbf{v}_t \in \mathcal{B}_d$, where $\mathcal{B}_d = \{\mathbf{v} \in \mathbb{R}^d : \|\mathbf{v}\| \leq 1\}$. Then play $\mathbf{w}_t$ by

$$\mathbf{w}_t = x_t \mathbf{v}_t$$

The interpretation of such strategy is $\mathcal{A}_{\mathbb{R}}$ as a magnitude learner and $\mathcal{A}_{\mathcal{B}_d}$ as a direction learner. Consequently, the regret of playing $\mathbf{w}_t$ is the related to regrets suffered by both learners as presented in Theorem 9. Hence allowing the extension of any algorithm operates on $\mathbb{R}$ to $\mathbb{R}^d$ without sacrificing regret guarantee by choosing appropriate direction learner $\mathcal{A}_{\mathcal{B}_d}$.

**Theorem 9.** *(Theorem 2 of Cutkosky & Orabona (2018)) Suppose $\mathcal{A}_{\mathcal{B}_d}$ obtains regret $R_T^{\mathcal{B}_d}(\mathbf{u})$ for any $\mathbf{u} \in \mathcal{B}_d$, $\mathcal{A}_{\mathbb{R}}$ obtains regret $R_T^{\mathbb{R}}(u)$ for any $u \in \mathbb{R}$. Let $\mathcal{A}_{\mathcal{B}_d}$ produce $\mathbf{v}_t$ in response to $\mathbf{g}_t$ and $\mathcal{A}_{\mathbb{R}}$ produce $x_t$ in response to $\langle \mathbf{g}_t, \mathbf{v}_t \rangle$. Then*

$$R_T(\mathbf{u}) = \sum_{t=1}^T \langle \mathbf{g}_t, \mathbf{w}_t - \mathbf{u} \rangle \leq R_T^{\mathbb{R}}(\|\mathbf{u}\|) + \|\mathbf{u}\| R_T^{\mathcal{B}_d}(\frac{\mathbf{u}}{\|\mathbf{u}\|})$$

We formally display Algorithm 4 as the dimension-free extension in the context of adversarial corruption in responding to $\tilde{\mathbf{g}}_t^c$ as a clipped version of $\tilde{\mathbf{g}}_t$ through gradient clipping in Equation (5) for some clipping threshold $0 < h_1 \leq \cdots \leq h_T$. Algorithm 4 is compatible with any algorithm $\mathcal{A}_{\mathbb{R}}$ operates on $\mathbb{R}$ and is referred as the magnitude learner. The direction learner $\mathcal{A}_{\mathcal{B}_d}$ is shown in Algorithm 5. We then present its $\mathcal{A}_{\mathbb{R}}$ dependent bound in Theorem 10.

---

**Algorithm 4** Dimension-free Robust Online Learning in Unconstrained Domain

---

**Require:** Time horizon $T$, $\tilde{\mathbf{g}}_t^c : \|\tilde{\mathbf{g}}_t^c\| \leq h_t$. $\mathcal{A}_{\mathbb{R}}$ operates on $\mathbb{R}$, $\mathcal{A}_{\mathcal{B}_d}$ operates on $\mathcal{B}_d$. Corruption parameter $k$, $\mathcal{A}_{\mathbb{R}}$ initialization parameter $\epsilon$.
1: **Initialize:**
     $\mathcal{A}_{\mathbb{R}}$ and Algorithm 5 as $\mathcal{A}_{\mathcal{B}_d}$.
2: **for** $t = 1$ to $T$ **do**
3:     Receive $x_t \in \mathbb{R}$ from $\mathcal{A}_{\mathbb{R}}$, $\mathbf{v}_t$ from $\mathcal{A}_{\mathcal{B}_d}$
4:     Play output $\mathbf{w}_t = x_t \mathbf{v}_t$, suffer loss $\langle \mathbf{g}_t, \mathbf{w}_t \rangle$
5:     Receive $\tilde{\mathbf{g}}_t^c, h_{t+1}$
6:     Send $z_t = \langle \tilde{\mathbf{g}}_t^c, \mathbf{v}_t \rangle, h_{t+1}$ to $\mathcal{A}_{\mathbb{R}}$, send $\tilde{\mathbf{g}}_t^c$ to $\mathcal{A}_{\mathcal{B}_d}$
7: **end for**

---

**Algorithm 5** Direction Learner: Online Subgradient Descent

---

**Require:** $\tilde{\mathbf{g}}_t^c, \mathbf{v}_1 = 0$
1: **for** $t = 1$ to $T$ **do**
2:     Output $\mathbf{v}_t$, receive $\tilde{\mathbf{g}}_t^c$
3:     Set learning rate $\eta_t = (\sum_{i=1}^t \|\tilde{\mathbf{g}}_i^c\|^2)^{-1/2}$
4:     Compute $\mathbf{v}_{t+1} \in \arg\min_{\mathbf{v}:\|\mathbf{v}\| \leq 1} \|\mathbf{v}_t - \eta_t \tilde{\mathbf{g}}_t^c\|$
5: **end for**

---

**Theorem 10.** *Suppose algorithm having access to $\tilde{\mathbf{g}}_t^c$ in receiving $\tilde{\mathbf{g}}_t$ as defined in Equation (5) with $0 < h_1 \leq h_2 \cdots \leq h_{T+1}$. Let $\mathcal{A}_{\mathbb{R}}$ be any algorithm operate on $\mathbb{R}$. Then Algorithm 4 runs on $\tilde{\mathbf{g}}_t^c$ guarantees:*

$$R_T(\mathbf{u}) \leq \sum_{t=1}^T \langle z_t, x_t - \|\mathbf{u}\| \rangle + \sum_{t=1}^T \|\mathbf{g}_t - \tilde{\mathbf{g}}_t^c\| (|x_t| + \|\mathbf{u}\|) + \frac{3\|\mathbf{u}\|}{2} h_T \sqrt{T}$$

*where $|z_t| \leq h_t$*

*Proof.* We begin with a convenience form of true regret in responding to $\tilde{\mathbf{g}}_t^c$:

$$R_T(\mathbf{u}) := \sum_{t=1}^T \langle \tilde{\mathbf{g}}_t^c, \mathbf{w}_t - \mathbf{u} \rangle + \sum_{t=1}^T \langle \mathbf{g}_t - \tilde{\mathbf{g}}_t^c, \mathbf{w}_t - \mathbf{u} \rangle$$

In the view of Theorem 9 for the first term:

$$R_T(\mathbf{u}) \leq \sum_{t=1}^T \langle z_t, x_t - \|\mathbf{u}\| \rangle + \|\mathbf{u}\| R_T^{\mathcal{B}_d}(\frac{\mathbf{u}}{\|\mathbf{u}\|}) + \sum_{t=1}^T \langle \mathbf{g}_t - \tilde{\mathbf{g}}_t^c, \mathbf{w}_t - \mathbf{u} \rangle$$

$$\leq \sum_{t=1}^T \langle z_t, x_t - \|\mathbf{u}\| \rangle + \sum_{t=1}^T \|\mathbf{g}_t - \tilde{\mathbf{g}}_t^c\| (\|x_t \mathbf{v}_t\| + \|\mathbf{u}\|) + \|\mathbf{u}\| R_T^{\mathcal{B}_d}(\frac{\mathbf{u}}{\|\mathbf{u}\|})$$

$$\leq \sum_{t=1}^{T} \langle z_t, x_t - \|\mathbf{u}\| \rangle + \sum_{t=1}^{T} \|\mathbf{g}_t - \tilde{\mathbf{g}}_t^c\| \left( |x_t| + \|\mathbf{u}\| \right) + \|\mathbf{u}\| R_T^{\mathcal{B}_d} \left( \frac{\mathbf{u}}{\|\mathbf{u}\|} \right)$$

where the last line is due to $\|\mathbf{v}_t\| \leq 1$. Moreover, $|z_t| = |\langle \tilde{\mathbf{g}}_t^c, \mathbf{v}_t \rangle| \leq \|\tilde{\mathbf{g}}_t^c\| \leq h_t$. And $R_T^{\mathcal{B}_d}(\mathbf{u}) \leq \frac{3\|\mathbf{u}\|}{2} \sqrt{\sum_{t=1}^{T} \|\tilde{\mathbf{g}}_t^c\|^2}$ is by following standard subgradient descent with Lipschitz adaptive learning rates for the second term (Theorem 4.14 of Orabona (2019)). Since $\|\tilde{g}_t^c\| \leq h_t \leq h_T$ we have $\|\mathbf{u}\| R_T^{\mathcal{B}_d}(\frac{\mathbf{u}}{\|\mathbf{u}\|}) \leq \frac{3\|\mathbf{u}\|}{2} h_T \sqrt{T}$ $\qquad\square$

**Theorem 11.** *Suppose $\mathbf{g}_t, \tilde{\mathbf{g}}_t$ satisfies assumptions in Equation (3) and (4). Algorithm 4 in response to $\tilde{\mathbf{g}}_t^c$ as defined in Equation (5) with $h_1 = \cdots, h_T = G$, by setting $\mathcal{A}_{\mathbb{R}}$ as Algorithm 1 with all parameters the same as that of Theorem 1. Then Algorithm 4 guarantees:*

$$R_T(\mathbf{u}) \leq \tilde{O}\left[ \epsilon + \|\mathbf{u}\| G \left( \sqrt{T} + k \right) \right]$$

*Proof.* By Theorem 10 and $h_T = G$

$$R_T(\mathbf{u}) \leq \sum_{t=1}^{T} \langle z_t, x_t - \|\mathbf{u}\| \rangle + \sum_{t=1}^{T} \|\mathbf{g}_t - \tilde{\mathbf{g}}_t^c\| \left( |x_t| + \|\mathbf{u}\| \right) + \frac{3\|\mathbf{u}\|}{2} G \sqrt{T}$$

$$\leq \sum_{t=1}^{T} \langle z_t, x_t - \|\mathbf{u}\| \rangle + \sum_{t=1}^{T} \|\mathbf{g}_t - \tilde{\mathbf{g}}_t\| \left( |x_t| + \|\mathbf{u}\| \right) + \frac{3\|\mathbf{u}\|}{2} G \sqrt{T}$$

due to $\mathbf{g}_t, \tilde{\mathbf{g}}_t$ satisfies assumptions in Equation (4)

$$\leq \sum_{t=1}^{T} \langle z_t, x_t - \|\mathbf{u}\| \rangle + kG \left( \max_t |x_t| + \|\mathbf{u}\| \right) + \frac{3\|\mathbf{u}\|}{2} G \sqrt{T}$$

In addition, $|z_t| \leq h_t = G$ is guaranteed by Theorem 10, hence apply Theorem 1 to the first two term

$$\leq \tilde{O}\left( \epsilon + \|\mathbf{u}\| G \left( \sqrt{T} + k \right) \right) + \frac{3\|\mathbf{u}\|}{2} G \sqrt{T}$$

$$= \tilde{O}\left( \epsilon + \|\mathbf{u}\| G \left( \sqrt{T} + k \right) \right)$$

$\qquad\square$

# E LOWER BOUNDS

In this section, we present two type of matching lower bounds to Theorem 1: Theorem 2 provides a lower bound for any comparator $u^* \in \mathbb{R}^d$ with arbitrary magnitude $D > 0$. Theorem 16 is a lower bound with log factors, which appears in unconstrained OCO upper bounds.

We begin by presenting a helper lemma that aids in the analysis of Theorem 2, followed by Lemmas required to proof to Theorem 2.

**Lemma 12.** *Suppose $z_1, z_2, \cdots, z_T \in \{-1, +1\}$ with equal probability. Then for every $t \in [T]$ for some $T \geq 1$.*

$$\mathbb{E}\left[ \sum_{t=1}^{T} sign \left( \sum_{i=1}^{T} z_i \right) z_t \right] \geq \sqrt{\frac{T}{16}}$$

*Proof.* Define $S_t = \sum_{i \in [T]: i \neq t} z_i$, by conditioning on $g_T \in \{-1, +1\}$:

$$2\mathbb{E}\left[ \text{sign} \left( \sum_{i=1}^{T} z_T \right) z_t \right] = \mathbb{E}\left[ \text{sign} \left( S_T + 1 \right) \right] - \left[ \text{sign} \left( S_T - 1 \right) \right]$$

$$= \sum_{k \in \{-T, -T+2, \cdots, T\}} \left( \text{sign}(k+1) - \text{sign}(k-1) \right) P(S_T = k)$$

We consider $T$ by cases: suppose $T$ is even, $\text{sign}(k+1) - \text{sign}(k-1) = 2$ when $k = 0$, and $\text{sign}(k+1) - \text{sign}(k-1) = 0$ otherwise. Thus applying $\binom{T}{T/2} \geq 2^{T-1}(T/2)^{-1/2}$

$$\mathbb{E}\left[\text{sign}\left(\sum_{i=1}^{T} z_i\right) z_T\right] = P(S_T = 0) = \binom{T}{T/2} 2^{-T} \geq 2^{-1}(T/2)^{-1/2} = \sqrt{\frac{1}{2T}}$$

Similarly if $T$ is odd, by symmetry to $S_T = \pm 1$:

$$\mathbb{E}\left[\text{sign}\left(\sum_{i=1}^{T} z_i\right) z_T\right] = \frac{1}{2}\left(P(S_T = -1) + P(S_T = 1)\right)$$

$$= \binom{T}{(T+1)/2} 2^{-T}$$

Define $T' = T - 1$ thus $T'$ is even

$$= \frac{T'!}{\left(\frac{T'}{2}\right)!\left(\frac{T'}{2}\right)!} \cdot \frac{\left(\frac{T'}{2}\right)!\left(\frac{T'}{2}\right)!}{\left(\frac{T'+2}{2}\right)!\left(\frac{T'}{2}\right)!} \cdot \frac{(T'+1)!}{(T')!} 2^{-(T'+1)}$$

$$= \binom{T'}{T'/2} \frac{T'+1}{\frac{T'+2}{2} + \frac{T'}{2}} 2^{-(T'+1)}$$

$$\geq 2^{T'-1}\left(\frac{T'}{2}\right)^{-1/2} \frac{T'+1}{T'+2} 2^{-(T'+1)}$$

$$\geq \frac{1}{8T'} = \frac{1}{8(T-1)} \geq \frac{1}{16T}$$

Thus combining two cases:

$$\mathbb{E}\left[\text{sign}\left(\sum_{i=1}^{T} z_i\right) z_T\right] \geq \frac{1}{16T}$$

Due to symmetry, $S_t$ has the same distribution $\forall t \in [T]$:

$$\mathbb{E}\left[\text{sign}\left(\sum_{i=1}^{T} z_T\right) z_t\right] = \mathbb{E}\left[\text{sign}\left(\sum_{i=1}^{T} z_i\right) z_T\right], \quad \forall t \in [T]$$

Thus

$$\mathbb{E}\left[\sum_{t=1}^{T} \text{sign}\left(\sum_{i=1}^{T} z_i\right) z_t\right] = T\mathbb{E}\left[\text{sign}\left(\sum_{i=1}^{T} z_i\right) z_T\right] \geq \sqrt{\frac{T}{16}}$$

$\square$

**Theorem 2.** *For every $D > 0$, there exists a comparator $u^* \in \mathbb{R}^d$ such that $\|u^*\| = D$, $\tilde{g}_1, \cdots, \tilde{g}_T$ and $g_1, \cdots, g_T$ such that $\|g_t\|, \|\tilde{g}_t\| \leq 1$, $\sum_{t=1}^{T} \mathbb{1}\{\tilde{g}_t \neq g_t\} = k$:*

$$\sum_{t=1}^{T} \langle g_t, w_t - u^* \rangle \geq \Omega\left[\|u^*\|\left(\sqrt{T} + k\right)\right]$$

*Proof.* Consider the following random sequence: $z_{k+1}, z_{k+2}, \cdots, z_T \in \{-1, +1\}$ with equal probability and $z_1 = \cdots, z_k = \text{sign}(\sum_{t=k+1}^{T} z_t)$. And $\tilde{z}_1 = \cdots = \tilde{z}_k = 0$ and $\tilde{z}a_t = z_t, \forall t \geq k + 1$. Let $q \in \mathbb{R}^n$ be any unity vector. Suppose $g_t = z_t q, \tilde{g}_t = \tilde{z}_t q, \forall t \in T$. Select $u^* = -D \text{sign}(\sum_{t=k+1}^{T} g_t)q$. Thus:

$$\mathbb{E}[R_T(u^*)] = \mathbb{E}\left[\sum_{t=1}^{T} \langle g_t, w_t - u \rangle\right]$$

$$= \sum_{t=1}^{T} \mathbb{E}\left[\langle g_t, w_t\rangle\right] - \mathbb{E}\left[\sum_{t=1}^{k}\langle g_t, u\rangle\right] - \sum_{t=k+1}^{T}\mathbb{E}\left[\langle g_t, u\rangle\right]$$

$$= \sum_{t=1}^{T}\mathbb{E}\left[\langle \mathbb{E}_t[z_t]q, w_t\rangle\right] + Dk + D\sum_{t=k+1}^{T}\mathbb{E}\left[z_t \operatorname{sign}\left(\sum_{t=k+1}^{T}z_t\right)\right]$$

$$= Dk + D\sum_{t=k+1}^{T}\mathbb{E}\left[z_t \operatorname{sign}\left(\sum_{t=k+1}^{T}z_t\right)\right]$$

by Lemme 12

$$\geq D\left(k + \sqrt{\frac{T-k}{16}}\right) = \Omega(\|u^*\|(k + \sqrt{T}))$$

$\square$

The second lower bound in Theorem 16 has a matching log factors by uses the definition of "regret at the origin" of an online learning algorithm, formalized as:

$$R_T(0) = \sum_{t=1}^{T}\langle g_t, w_t - 0\rangle \leq \epsilon \tag{18}$$

This condition implies that an algorithm maintaining small $\epsilon$ is inherently conservative: it will perform well if the comparator is close to the origin, but this behavior may come at the cost of performing poorly if the comparator is far from the origin. Before presenting the analysis to Theorem 16, we first list previously established result on properties of iterates $w_t$ produced by any algorithm has constant regret guarantee at the origin as defined in Equation (18). Lemma 13 was originally appeared in Cutkosky (2018) then being re-interpreted by Orabona (2019). Lemma 14 from Zhang & Cutkosky (2022).

**Lemma 13** (Theorem 5.11 of Orabona (2019)). *For any OLO algorithm suffers constant regret at the origin (Equation (18)) and $|g_t| \leq 1$, there exist $\beta_t \in R^d$ such that $\|\beta_t\| \leq 1$ and*

$$w_t = \beta_t\left(\epsilon - \sum_{i=1}^{t-1}\langle g_i, w_i\rangle\right)$$

*for all $t \in [T]$.*

**Lemma 14** (Lemma 8 of Zhang & Cutkosky (2022): Unconstrained OLO Iterate Growth). *Suppose assumptions in Lemma 13 is satisfied. Then for every $t \in [T]$, $\|w_t\| \leq \epsilon 2^{t-1}$.*

We first derive an lower bound for algorithms satisfies assumption in Lemma 13. The construction was originally appeared in Theorem 5.12 from Orabona (2019). Finally, the lower bound in the context of adversarial corruptions is presented in Theorem 2.

**Lemma 15** (Unconstrained OLO Lower Bound). *Suppose assumptions in Lemma 13 is satisfied, then set $g_t = [g_{t,1}, 0, \cdots, 0]$, $g_{t,1} = g = 1$ for all $t \in [T]$. Then there exists an $u^* \in \mathbb{R}^d$ such that $\|u^*\| = 2\epsilon e^T$, and*

$$\sum_{t=1}^{T}\langle g_t, w_t - u^*\rangle \geq \epsilon + \|u^*\|\sqrt{\frac{T}{30}\ln\left(1 + \frac{\|u^*\|^2 T}{2\epsilon^2}\right)}$$

*Proof.* Let $r_t = -\sum_{i=1}^{t}\langle g_i, w_i\rangle$. Then

$$\epsilon - \sum_{t=1}^{T}\langle g_t, w_t\rangle = \epsilon + r_{T-1} - \langle g_T, w_T\rangle$$

by Lemma 13, there exists some $\beta_T : \|\beta_T\| \leq 1$

$$= \epsilon + r_{T-1} - \langle g_T, \beta_T\rangle(\epsilon + r_{T-1})$$

$$= (1 - \langle g_t, \beta_t \rangle)(\epsilon + r_{T-1})$$

Then recursively expand $r_{T-1}, r_{T-2}, \cdots, r_1$ with Lemma 13, then for some $\beta_t : \|\beta_t\| \le 1$

$$\epsilon - \sum_{t=1}^{T} \langle g_t, w_t \rangle = \epsilon \prod_{t=1}^{T} (1 - \langle g_t, \beta_t \rangle)$$

Hence

$$\epsilon - \sum_{t=1}^{T} \langle g_t, w_t \rangle \le \epsilon \prod_{t=1}^{T} \max_{\|\beta_t\| \le 1} (1 - \langle g_t, \beta_t \rangle) = \epsilon \prod_{t=1}^{T} (1 + |g|) = \epsilon \left(1 + \frac{|g|^2 T}{T}\right)^T \le \epsilon \exp\left(|g|^2 T\right)$$

where we used inequality $(1 + \frac{x}{n})^n \le e^x$ by setting $n = T, x = |g|^2 T$ for the last step. Rearrange above equation, we have

$$\sum_{t=1}^{T} \langle g_t, w_t \rangle - \epsilon \ge -\epsilon \exp\left(|g|^2 T\right) = -\epsilon \exp\left(\frac{|\sum_{t=1}^{T} g_{t,1}|^2}{T}\right) = -f(-\sum_{t=1}^{T} g_{t,1})$$

where $f(x) = \epsilon \exp(\frac{x^2}{T})$, by Theorem 27 part 1, we have $f(x) = f^{**}(x)$. Then by the definition of double conjugate $f^{**}$,

$$\sum_{t=1}^{T} \langle g_t, w_t \rangle - \epsilon \ge -f^{**}(-\sum_{t=1}^{T} g_{t,1}) = -\left(\sup_{u_1 \in \mathbb{R}} \langle -\sum_{t=1}^{T} g_{t,1}, u_1 \rangle - f^*(u_1)\right) \tag{19}$$

By Theorem 27 part 2, the supreme is achieve at

$$u_1^* = \nabla f(-\sum_{t=1}^{T} g_{t,1}) = \frac{2\epsilon}{T} \left(\sum_{t=1}^{T} g_{t,1}\right) \exp\left(\frac{\left(\sum_{t=1}^{T} g_{t,1}\right)^2}{T}\right) = 2\epsilon e^T$$

Substitute $u_1^*$ and set $u^* = [u_1^*, 0, \cdots, 0]$, then Equation (19) becomes:

$$\sum_{t=1}^{T} \langle g_t, w_t \rangle - \epsilon \ge \sum_{t=1}^{T} \langle g_{t,1}, u_1^* \rangle + f^*(u_1^*) = \sum_{t=1}^{T} \langle g_t, u^* \rangle + f^*(u_1^*)$$

Rearrange we have

$$\sum_{t=1}^{T} \langle g_t, w_t - u^* \rangle \ge \epsilon + f^*(u_1^*) \tag{20}$$

It remains to obtain a lower bound to $f^*(u_1^*)$. By Lemma 29 and Lemma 28, we have

$$f^*(u_1^*) = \sqrt{\frac{T}{2}} |u_1^*| \left(\sqrt{W\left(\frac{T|u_1^*|^2}{2\epsilon^2}\right)} - \frac{1}{\sqrt{W\left(\frac{T|u_1^*|^2}{2\epsilon^2}\right)}}\right)$$

$$\ge \sqrt{\frac{T}{2}} |u_1^*| \left(\sqrt{0.6 \ln\left(1 + \frac{T|u_1^*|^2}{2\epsilon^2}\right)} - \frac{1}{\sqrt{0.6 \ln\left(1 + \frac{T|u_1^*|^2}{2\epsilon^2}\right)}}\right)$$

Notice that $0.6 \ln\left(1 + \frac{T|u_1^*|^2}{2\epsilon^2}\right) = 0.6 \ln(1 + 2\exp(T)^2 T) > 1.5$, hence by Lemma 30

$$\ge \sqrt{\frac{T}{2}} |u_1^*| \sqrt{\frac{0.2}{3} \ln\left(1 + \frac{T|u_1^*|^2}{2\epsilon^2}\right)}$$

$$= |u_1^*| \sqrt{\frac{T}{30} \ln\left(1 + \frac{T|u_1^*|^2}{2\epsilon^2}\right)}$$

Substitute the lower bound to $f^*(u_1^*)$ to Equation (20)

$$\sum_{t=1}^{T}\langle g_t, w_t - u^*\rangle \geq \epsilon + |u_1^*|\sqrt{\frac{T}{30}\ln\left(1 + \frac{|u_1^*|^2 T}{2\epsilon^2}\right)} = \epsilon + \|u^*\|\sqrt{\frac{T}{30}\ln\left(1 + \frac{\|u^*\|^2 T}{2\epsilon^2}\right)}$$

$\square$

**Theorem 16.** *For any algorithm that maintains Equation (18) for some $\epsilon > 0$, there exists a sequence of $\tilde{g}_1, \cdots, \tilde{g}_T$ and $g_1, \cdots, g_T$ such that $\|g_t\|, \|\tilde{g}_t\| \leq 1$, $\sum_{t=1}^{T} \mathbb{1}\{\tilde{g}_t \neq g_t\} = k$, and a $u^* \in \mathbb{R}^d$ such that*

$$\sum_{t=1}^{T}\langle g_t, w_t - u^*\rangle \geq \tilde{\Omega}\left[\epsilon + \|u^*\|\left(\sqrt{T} + k\right)\right]$$

*Proof.* the proof strategy is that algorithm with regret guarantee as shown in Equation (18) attains a matching lower bound $\tilde{\Omega}(\epsilon + \|u\|\sqrt{T})$ in responding to $\mathbf{g}_t$ as shown in Lemma 15. The by reversing the direction of exactly $k$ gradients by taking account into the growth behavior of $w_t$ (Lemma 14) and a particular hard comparator $u^*$ constructed in Lemma 15, we can show regrets during those rounds builds up linearly. Let $\tilde{g}_1, \cdots, \tilde{g}_T$, where $\|\tilde{g}_t\| \leq 1$ as defined in Lemma 15 and suppose algorithm operates on those gradients. Let $S$ be the index set $S = \{t \in [T] : g_t \neq \tilde{g}_t\}$. Then by the lower bound presented in Lemma 15

$$\sum_{t=1}^{T}\langle g_t, w_t - u^*\rangle = \sum_{t=1}^{T}\langle \tilde{g}_t, w_t - u^*\rangle + \sum_{t=1}^{T}\langle g_t - \tilde{g}_t, w_t - u^*\rangle$$

$$\geq \tilde{\Omega}(\epsilon + \|u^*\|\sqrt{T}) + \sum_{t\in S}\langle g_t - \tilde{g}_t, w_t - u^*\rangle$$

for some $u^* \in \mathbb{R}^d$ and $\|u^*\| = 2\epsilon e^T$. For $t \in S$, define $g_t$ as follows

$$g_t = \tilde{g}_t - \frac{u^*}{\|u^*\|}$$

Then

$$\sum_{t=1}^{T}\langle g_t, w_t - u^*\rangle \geq \tilde{\Omega}(\epsilon + \|u^*\|\sqrt{T}) + \sum_{t\in S}\langle -\frac{u^*}{\|u^*\|}, w_t\rangle + \sum_{t\in S}\langle \frac{u^*}{\|u^*\|}, u^*\rangle$$

$$\geq \tilde{\Omega}(\epsilon + \|u^*\|\sqrt{T}) - \sum_{t\in S}\|w_t\| + k\|u^*\|$$

Finally, By Lemma 14 $\|w_t\| \leq \epsilon 2^{t-1}$. Hence $\|w_t\| \leq \frac{1}{2}\|u^*\|$

$$\sum_{t=1}^{T}\langle g_t, w_t - u^*\rangle \geq \tilde{\Omega}(\epsilon + \|u^*\|\sqrt{T}) - \frac{k}{2}\|u^*\| + k\|u^*\| = \tilde{\Omega}\left(\epsilon + \|u^*\|\left(\sqrt{T} + k\right)\right)$$

$\square$

# F ADAPTIVE THRESHOLDING

In this section, we formalize the adaptive thresholding and clipping mechanism, namely FILTER, summarized in Section 5.1. This mechanism relies on prior knowledge of big corrupted gradients numbers which is naturally restricted by corruption model in Equation (3). We present this result as Lemma 17, followed FILTER as Algorithm 6 and its property in Lemma 18.

**Lemma 17.** *For $g_1, \cdots, g_T$ and $\tilde{g}_1, \cdots, \tilde{g}_T$ that satisfies Equation (3), then there are at most $k$ number of $\tilde{g}_t$ such that $\|\tilde{g}_t\| \geq 2G$.*

*Proof.* By definition of $\mathcal{B} = \{t \in [T] : \|g_t - \tilde{g}_t\| > G\}$:

$$\begin{aligned}
\mathcal{B} &:= \{t \in [T] : \|g_t - \tilde{g}_t\| > G\} \\
&= \{t \in [T] : \|g_t - \tilde{g}_t\| > G, \|g_t\| < G\} \cup \{t \in [T] : \|g_t - \tilde{g}_t\| > G, \|g_t\| = G\} \\
&\supseteq \{t \in [T] : \|g_t - \tilde{g}_t\| > G, g_t = G \cdot \text{sign}(\tilde{g}_t)\} \\
&= \{t \in [T] : \|G - \|\tilde{g}_t\|\| > G\} \\
&= \{t \in [T] : \|\tilde{g}_t\| > 2G\}
\end{aligned}$$

Finally, due to Equation (3), $k := |\mathcal{B}| \geq |\{t \in [T] : \|\tilde{g}_t\| > 2G\}|$. $\qquad\qquad\square$

---

**Algorithm 6** FILTER: $k$-lag Thresholding and Gradient Clipping

---

**Require:** Corruption parameter $k$, Initial Lipschitz guess: $\tau = \tau_G > 0$.
 1: **Initialize:**
        Filter threshold $h_1 = \tau$, Check point $h = h_1$, Counter: $n = 0$, $\mathcal{P} = \{\}$
 2: **for** $t = 1$ to $T$ **do**
 3:     Receive $\tilde{g}_t$
 4:     **if** $\|\tilde{g}_t\| > h$ **then**
 5:         Set $\tilde{g}_t^c = \frac{\tilde{g}_t}{\|\tilde{g}_t\|} h_t$, update counter: $n = n + 1$
 6:         Update threshold $h_{t+1} = h_t + \frac{1}{k+1} h$
 7:         **if** $n = k$ **then**
 8:             Update Check point $h = h_{t+1}$, reset counter: $n = 0$
 9:         **end if**
10:     **else**
11:         Set $\tilde{g}_t^c = \tilde{g}_t$, register rounds $\mathcal{P} = \mathcal{P} \cup t$
12:         Maintain threshold $h_{t+1} = h_t$
13:     **end if**
14:     Output $\tilde{g}_t^c, h_{t+1}$
15: **end for**

---

We display some convenience property of Algorithm FILTER, notice all quantities apart from $h_t$ are for assisting analysis only

**Lemma 18.** *(Algorithm 6 property) Suppose $g_t, \tilde{g}_t$ satisfies Equation (3), and Algorithm 6 receives $\tilde{g}_t$, then its per iteration outputs $\tilde{g}_t^c, h_{t+1}$ satisfies:*

*(1) $h_{t+1} = h_t, \forall t \in \mathcal{P} = \{t \in [T] : \tilde{g}_t^c = \tilde{g}_t\}$*

*(2) $\|\tilde{g}_t^c\| \leq h_t, \forall t \in [T]$*

*(3) $\tau = h_1 \leq h_2 \leq \cdots \leq h_{T+1} \leq \max(\tau, 8G)$*

*(4) $|\mathcal{P}| \geq T - (k+1) \max\left(\lceil \log_2 \frac{8G}{\tau} \rceil, 1\right)$*

*(5) $h_{t+1}/(h_{t+1} - h_t) \leq 2(k+1), \forall t \notin \mathcal{P}$*

*Proof.* We show each property in turns.

(1) guaranteed by algorithm line 11-12.

(2) either line 4 or line 11 is evoked to compute $\tilde{g}_t^c$.

(3) $h_t$ being non-decreasing sequence and $h_t = \tau$ is by construction. Hence it remains to show an upperbound to $h_t, \forall t \in [T+1]$. The key to this proof is there are at most $k$ number of $\tilde{g}_t$ such that $\|\tilde{g}_t\| \geq 2G$ gaurateed by Equation (3) (See Lemma 17).

In the case where initial value of $\tau \geq 2G$, then the check point $h$ never doubled since each time of doubling requires $k + 1$ number of $\|\tilde{g}_t\|$ exceeds current one. (by line 4-9)

Now, we consider $\tau < 2G$, where doubling of check point $h$ was evoked at least once (by evoking line 8) with initial value $\tau$, then $h \in \{\tau, 2\tau, 2^2\tau, \cdots, 2^N\tau\}$ for some $N \in [T]$,

where $N$ is the number of time line 8 was evoked. Then $2^N \tau \le h_t \le 2^{N+1}\tau, \forall t \in [t^*, T+1]$ where $t^* \le T$ is the last time step where $h$ was doubled.

On the other hand, $h \in \{2^{N-1}\tau, 2^N\tau\}$ at some period of time. This means during this time interval at least $k+1$ number of $\tilde{g}_t$ such that $\|\tilde{g}_t\| \ge 2^{N-1}\tau$ were observed thus have triggered line 8. Thus $2^{N-1}\tau \le 2G$, $N+1 \le \log_2 \frac{8G}{\tau}$.

Combining both conclusions from above $h_t \le 2^{N+1}\tau \le 8G, \forall t \in [t^*, T+1]$. Moreover, $h_t$ is non-decreasing, and we complete the proof.

(4) $|\mathcal{P}|$ is associated with the number of time in which check point $h$ doubled. By the proof to property (3) that $2^{N-1}\tau \le 2G$, thus $N \le \max\left(\lceil \log_2 \frac{4G}{\tau} \rceil, 0\right)$ as an upper bound that the number of $h$ being doubled.

For $t \le t^*$, each doubling requires exactly $k+1$ number of $\tilde{g}_t$ being clipped. Thus there were $(k+1)\max\left(\lceil \log_2 \frac{4G}{\tau} \rceil, 0\right)$ number of rounds not being register to $\mathcal{P}$. For $t > t^*$, there were less than $(k+1)$ number of $\tilde{g}_t$ not being registered into $\mathcal{P}$, otherwise threshold would have been doubled. Thus

$$|\mathcal{P}| \le (k+1)\max\left(\left\lceil \log_2 \frac{4G}{\tau} \right\rceil, 0\right) + (k+1) = (k+1)\max\left(\left\lceil \log_2 \frac{8G}{\tau} \right\rceil, 1\right)$$

(5) For $t \notin \mathcal{P}$, $h_{t+1} = (1 + \frac{n+1}{k+1})h$, $h_t = (1 + \frac{n}{k+1})h$, for some $n \in [k]$ and for some $h \in \{\tau, 2\tau, 2^2\tau, \cdots, 2^{t'}\tau\}$. Hence

$$\frac{h_{t+1}}{h_{t+1} - h_t} = \frac{1 + \frac{n+1}{k+1}}{\frac{n+1}{k+1} - \frac{n}{k+1}} = 2 + k + n \le 2(k+1)$$

$\square$

## G  ADAPTIVE TRACKING

We introduce TRACKER, an adaptive mechanism for estimating $\max_t |w_t|$. as shown in Algorithm 7. TRACKER maintains thresholds $z_t$ in which doubles whenever $\|w_t\| > z_t$. The properties of TRACKER is displayed in Lemma 19.

---

**Algorithm 7** TRACKER: track the magnitude of $w_t$

**Require:** Initial magnitude guess: $\tau = \tau_D > 0$.
1: **Initialize:**
     Filter threshold $z_1 = \tau$, (Counter, Set): $(n = 0, \mathcal{T}_n = \{\})$, Check point $t_0 = 1$
2: **for** $t = 1$ to $T$ **do**
3:     Receive $w_t$
4:     **if** $\|w_t\| > z_t$ **then**
5:         Double: $z_{t+1} = 2z_t$
6:         Update counter $n = n + 1$
7:         Add a new checkpoint: $t_n = t$, add a new set $\mathcal{T}_n = \{\}$)
8:     **else**
9:         Maintain: $z_{t+1} = z_t$
10:     **end if**
11:     Register round $\mathcal{T}_n \leftarrow \mathcal{T}_n \cup t$
12: **end for**

---

**Lemma 19.** *(Algorithm 7 property) Algorithm 7 guarantees*

*(1)* $[T]$ *is partitioned by* $\mathcal{T}_0, \mathcal{T}_1, \mathcal{T}_2, \cdots, \mathcal{T}_N$, *for some* $N \le T$

*(2)* $\tau = z_t = z_{t+1}, |w_t| \le \tau, \forall t \in \mathcal{T}_0$

*(3)* $z_{t+1} = 2z_t$ *iff* $t = t_n, n \in [N]$, $z_{t+1} = z_t$ *otherwise*

*(4)* $\|w_t\| \le 2\|w_{t_n}\|, \forall t \in \mathcal{T}_n, n \in [N]$

*(5)* $\tau = z_1 \le z_2 \le \cdots \le z_{T+1} \le \max(\tau, 2\max_t |w_t|)$

*Proof.* We show each property in turns.

(1) partition property can be seen by in the initialization of $n = 0$ with increment of 1 (line 6) and whenever counter $n$ updates a new set $\mathcal{T}_n$ is created (line 7). And $\forall t \in [T]$ is assigned to $\mathcal{T}_n$ for some $n \ge 0$ (line 11).

(2) For the time period of $n = 0$, line 4 was never executed.

(3) As $n \ge 1$: $z_{t_n+1} = 2z_{t_n}$ and $\|w_{t_n}\| > z_{t_n}$ when line 5 was evoked. otherwise $z_{t+1} = z_t$ as in line 9 where $t_n \ne t$.

(4) By construction $\mathcal{T}_n = \{t_n, t_n + 1, \cdots, t_{n+1} - 1\}, \forall n \in [N-1], \mathcal{T}_N = \{t_N, \cdots, T\}$. When $t = t_n$, the inequality holds. Thus we consider $\forall t \in \mathcal{T}_n \setminus \{t_n\}$, line 9 was triggered, hence $z_{t+1} = z_t = z_{t_n+1}$ and $\|w_t\| \le z_t$. On the other hand, by property (2) $z_{t_n+1} = 2z_{t_n}$ and $\|w_{t_n}\| > z_{t_n}$. Thus

$$2\|w_{t_n}\| > 2z_{t_n} = z_{t_n+1} = z_t \ge \|w_t\|, \quad \forall t \in \mathcal{T}_n \setminus \{t_n\}$$

(5) since $z_1 = \tau$ and $z_{t+1}$ is either through line 5 (double) or line 9 (maintain). Thus non-decreasing property holds.

Suppose line 5 was never executed, then $z_{T+1} = z_1 = \tau$. Now we consider line 5 was executed at least once. Let $t^* \in [T]$ be the last time step in which line 5 was executed. Thus

$$z_{T+1} = z_T = \cdots = z_{t^*+1} = 2z_{t^*} < 2\|w_{t^*}\|$$

a further upper bound is $z_t \le 2\max_t \|w_t\|$ for $t \in [t^* + 1, T + 1]$, combing with $z_t$ being non-decreasing, we complete the proof.

$\square$

## H  EPIGRAPH-BASED REGULARIZATION AND OPTIMISM

In this section, we present bound $R_T^{\mathcal{A}}(u)$ as defined in Equation (14) as Theorem 23. This bound is achieved by a combination of a recently developed Epigraph-based regularization Cutkosky & Mhammedi (2024) and optimistic online learning as derived in Theorem 7, Appendix B. In this section, all quantities are from Algorithm 2.

We begin with introducing the necessity of such combination by the decomposition of $R_T^{\mathcal{A}}(u)$ by taking advantage of $r_t$ being convex:

$$R_T^{\mathcal{A}}(u) := \sum_{t=1}^{T} \langle \tilde{g}_t^c, w_t - u \rangle + a_t \psi(w_t) - a_t \psi(u) + r_t(w_t) - r_t(u)$$

$$\le \sum_{t=1}^{T} \langle \tilde{g}_t^c + \nabla r_t(w_t), w_t - u \rangle + a_t \psi(w_t) - a_t \psi(u) \tag{21}$$

we abstain from treating $\alpha_t \psi(w)$ the same way as $r_t(w)$, since the linearization $\alpha_t \nabla \psi(w)$ is equivalent of learning a composite loss $w \mapsto \tilde{g}_t + \nabla r_t(w_t) + a_t \nabla \psi(w_t)$ as introduced in Appendix B. Thus, even through the optimistic reduction, Theorem 7 indicates the result will have linear dependence on $\max_t \alpha_t |\nabla \psi(w_t)| = O(\max_t |w_t|)$. Thus an alternative treatment needed to control $\sum_t a_t \psi(w_t) - a_t \psi(u)$.

Epigraph-based Regularization is the appropriate tool to keep $\sum_t a_t \psi(w_t) - a_t \psi(u)$ being under control through a geometric reparameterization. If an algorithm outputs $(w_t, y_t) \in W = \{(w, y):$

$y \geq w^2\} \subseteq \mathbb{R}^2$. Then Equation (21) can be further bounded by sum of two regrets:

$$R_T^{\mathcal{A}}(u) \leq \underbrace{\sum_{t=1}^{T} \langle \tilde{g}_t^c + \nabla r_t(w_t), w_t - u \rangle}_{R_T^{\mathcal{A}_w}(u)} + \underbrace{\sum_{t=1}^{T} a_t(y_t - u^2)}_{R_T^{\mathcal{A}_y}(u)} \tag{22}$$

Due to $W$ is an epigraph of $w^2$, this method was referred as "epigraph-based" regularization. We consider two unconstrained learner: $\mathcal{A}_w$ in producing $\hat{w}_t \in \mathbb{R}$ and $\mathcal{A}_y$ in producing $\hat{y}$. Before we can see how this is linked with $R_T^{\mathcal{A}_w}(u), R_T^{\mathcal{A}_y}(u)$, we first present a useful definition.

**Definition 20.** *For the set* $W = \{(w, y) : y \geq w^2\} \subseteq \mathbb{R}^2$, *and arbitrary* $(w, y) \in W$ *and* $(\hat{w}, \hat{y}) \in \mathbb{R}^2$ *and some* $h_t, \gamma > 0$:

    *(1) norm:* $\|(w, y)\|_t = h_t^2 w^2 + \gamma^2 y^2$

    *(2) dual norm:* $\|(w, y)\|_{*,t} = \frac{w^2}{h_t^2} + \frac{y^2}{\gamma^2}$

    *(3) distance function of* $(\hat{w}, \hat{y})$ *to* $W$: $S_t((\hat{w}, \hat{y})) = \inf_{y \geq w^2} \|(w, y) - (\hat{w}, \hat{y})\|_t$

    *(4) subgradient at* $(\hat{w}, \hat{y})$: $\nabla S_t((\hat{w}, \hat{y})) = \left( \frac{h_t^2(\hat{w}-w)}{h_t^2(\hat{w}-w)^2 + \gamma^2(\hat{y}-y)^2}, \frac{\gamma^2(\hat{y}-y)}{h_t^2(\hat{w}-w)^2 + \gamma^2(\hat{y}-y)^2} \right)$

    *(5) projection map* $\Pi_W^t((\hat{w}, \hat{y})) = \arg\min_{(w,y) \in W} \|(w, y) - (\hat{w}, \hat{y})\|_t$

Roughly speaking, the black-box reduction in converting any unconstrained algorithm to operates on $W$ and enjoy the same regret guarantee of the unconstrained one (Cutkosky & Orabona, 2018) by projection $(w_t, y_t) = \Pi_W^t(\hat{w}_t, \hat{y}_t)$ and a gradient correction direction to avoid out of $W$ allows $R_T^{\mathcal{A}_w}(u) \leq \tilde{O}(|u|(h_T + |\nabla r_t|)\sqrt{T})$ and $R_T^{\mathcal{A}_y}(u) \leq \tilde{O}(|u|^2 \sqrt{\sum_t a_t^2})$ (also see Theorem 10 Cutkosky & Mhammedi (2024)). Those match the optimal unconstrained OCO rates.

However, $R_T^{\mathcal{A}_w}(u)$ might still not be satisfactory for our purpose since $|\nabla r_t|$ can be as large as $O(k)$ similarly as introduced in Section 4.1. Thus we choose $\mathcal{A}^w$ as a optimistic online learning algorithm that yields to $R_T^{\mathcal{A}_w}(u) \leq \tilde{O}(|u|h_T\sqrt{T} + |\nabla r_t|)$, and $\mathcal{A}^y$ being a standard unconstrained OCO with optimal rates will satisfy our need. Before presenting the analysis of $R_T^{\mathcal{A}}(u)$, we first introduce helper Lemmas:

**Lemma 21.** *In the same notation as Definition 20, if* $|g_t| \leq h_t$ *and* $\alpha_t \in [0, \gamma]$, *and* $(\delta_t^w, \delta_t^y) = \|(g_t, a_t)\|_{*,t} \nabla S_t((\hat{w}_t, \hat{y}_t))$ *then*

$$|\delta_t^w| \leq \sqrt{2} h_t, \qquad |\delta_t^y| \leq \sqrt{2} \gamma$$

*Proof.* Since $|g_t| \leq h_t$ and $\alpha_t \in [0, \gamma]$, $\|(g_t, a_t)\|_{*,t} \leq 2$. On the other hand $\|\nabla S_t((\hat{w}, \hat{y}))\|_{*,t} = 1$, and

$$\|(\delta_t^w, \delta_t^y)\|_{*,t} = \frac{|\delta_t^w|^2}{h_t^2} + \frac{|\delta_t^y|^2}{\gamma^2}$$

Thus

$$\frac{|\delta_t^w|^2}{h_t^2} + \frac{|\delta_t^y|^2}{\gamma^2} \leq 2$$

This implies both $\frac{|\delta_t^w|^2}{h_t^2} \leq 2$ and $\frac{|\delta_t^y|^2}{\gamma^2} \leq 2$. $\qquad\qquad\square$

**Lemma 22** (from Cutkosky & Mhammedi (2024)). *For* $0 < \gamma_1 \leq \gamma_2 \leq \cdots \leq \gamma_{T+1}$ *and* $\gamma_0 \geq 0$, *define*

$$\alpha_t = \gamma_0 \cdot \frac{(\gamma_{t+1} - \gamma_t)/\gamma_{t+1}}{1 + \sum_{i=1}^{t}(\gamma_{i+1} - \gamma_i)/\gamma_{i+1}}$$

*Then*

$$\sum_{t=1}^{T} \alpha_t \leq \gamma_0 \ln\left(\ln\left(1 + \frac{\gamma_{T+1}}{\gamma_1}\right)\right)$$

**Theorem 23.** *Suppose $g_t, \tilde{g}_t$ satisfies assumptions in Equation (3) and (4), and having access to $\tilde{g}_t^c$ as defined in Equation (5) with $h_t$ provided by* FILTER *(Algorithm 6). with $\alpha = \epsilon/c, \gamma_\alpha = \gamma_\beta = \gamma/2$, for some $\epsilon, c, \gamma, \tau_G, \tau_D > 0$*

$$R_T^{\mathcal{A}}(u) \leq \tilde{O}\left(\epsilon + |u|\max(\tau_G, G)\sqrt{T} + |u|c + |u|^2\gamma\right)$$

*In addition, the produced iterate satisfies $\max_t |w_t| \leq \frac{\epsilon}{2G}2^T$*

*Proof.* Algorithm 2 denote $\hat{w}_t, \hat{y}_t$ as outputs from some unconstrained learner and $w_t, y_t$ being their projection on $W$. We begin our analysis from Equation (22):

$$R_T^{\mathcal{A}}(u) \leq \sum_{t=1}^T \langle \tilde{g}_t^c + \nabla r_t(w_t), w_t - u\rangle + \sum_{t=1}^T a_t(y_t - \psi(u))$$

By Cutkosky & Orabona (2018) Theorem 3

$$\leq \underbrace{\sum_{t=1}^T \langle \tilde{g}_t^c + \nabla r_t(w_t) + \delta_t^w, \hat{w}_t - u\rangle}_{R_T^{\mathcal{A}_w}(u)} + \underbrace{\sum_{t=1}^T (a_t + \delta_t^y)(\hat{y}_t - \psi(u))}_{R_T^{\mathcal{A}_y}(u)} \quad (23)$$

Since $\gamma_\alpha = \frac{\gamma}{2}, \gamma_\beta = \frac{\gamma}{2}, a_t = \alpha_t + \beta_t \leq \gamma$. Thus, by Lemma 21, $|\tilde{g}_t^c + \delta_t^w| \leq h_t + \sqrt{2}(h_t + c\ln T) \leq 3(h_t + c\ln T)$ and $|a_t + \delta_t^y| \leq \gamma + \sqrt{2}\gamma \leq 3\gamma$. If both $\mathcal{A}_w, \mathcal{A}_y$ are standard unconstrained OCO algorithm, Theorem 10 of Cutkosky & Mhammedi (2024) implies

$$R_T^{\mathcal{A}_w}(u) \leq \tilde{O}\left(\epsilon + |u|(h_T + c)\sqrt{T}\right), \quad R_T^{\mathcal{A}_y}(u) \leq \tilde{O}\left(\epsilon + |u|^2\sqrt{\gamma^2 + \gamma\sum_{t=1}^T a_t}\right)$$

However, $\mathcal{A}_w$ is indeed an optimistic online learning algorithm by leveraging the known structure of $\nabla r_t(w_t)$ and $\delta_t^w$, a better bound in $R_T^{\mathcal{A}_w}(u)$ can be obtained by Theorem 7, which implies Algorithm 2 guarantees the following by setting $g_t \leftarrow \frac{1}{2}\tilde{g}_t^c, r_t \leftarrow \frac{1}{2}(\nabla r_t(w_t) + \delta_t^w)$ and $G_t = \frac{1}{2}h_t, H_t = \frac{3}{2}(h_{t+1} + c\ln T)$:

$$R_T^{\mathcal{A}_w}(u) \leq \tilde{O}\left(\epsilon + |u|h_T\sqrt{T} + |u|(h_T + c)\right) = \tilde{O}\left(\epsilon + |u|h_T\sqrt{T} + |u|c\right)$$

Thus, combing with $R_T^{\mathcal{A}_y}(u)$, we can bound Equation (23):

$$R_T^{\mathcal{A}}(u) \leq \sum_{t=1}^T \langle \tilde{g}_t^c + \delta_t^w + \nabla r_t(w_t), w_t - u\rangle + \sum_{t=1}^T (a_t + \delta_t^y)(y_t - \psi(u))$$

$$= \tilde{O}\left(\epsilon + |u|h_T\sqrt{T} + |u|c + |u|^2\sqrt{\gamma^2 + \gamma\sum_{t=1}^T a_t}\right)$$

since $\sum_t a_t = \sum_t \alpha_t + \sum_t \beta_t$, where $\alpha_t, \beta_t$ are defined in Algorithm 2 line 8. Thus, we apply Lemma 22 for each summand with appropriate substitutions

$$\leq \tilde{O}\left(\epsilon + |u|h_T\sqrt{T} + |u|c + |u|^2\sqrt{\gamma^2 + \gamma\left(\frac{\gamma}{2}\ln\left(\ln\left(1 + \frac{h_{T+1}}{h_1}\right)\right) + \frac{\gamma}{2}\ln\left(\ln\left(1 + \frac{z_{T+1}}{z_1}\right)\right)\right)}\right)$$

by Lemma 18 (3): $h_1 = \tau_G, h_T, h_{T+1} \leq \max(\tau_G, 8G)$, similarly by Lemma 19 (5): $z_1 = \tau_D, z_{T+1} \leq \max(\tau_D, 2\max_t |w_t|)$

$$\leq \tilde{O}\left(\epsilon + |u|\max(\tau_G, G)\sqrt{T} + |u|c\right)$$

$$+ \tilde{O}\left(|u|^2\gamma\sqrt{1 + \ln\left(\ln\left(1 + \max(1, \frac{G}{\tau_G})\right)\right) + \ln\left(\ln\left(1 + \max(1, \frac{\max_t |w_t|}{\tau_D})\right)\right)}\right)$$

By Lemma 8 of Zhang & Cutkosky (2022), $\max_t |w_t| \leq \frac{\epsilon}{2G}2^T$, thus the double logarithm in $\max_t |w_t|$ is at worst $O(\ln T)$

$$= \tilde{O}\left(\epsilon + |u|\max(\tau_G, G)\sqrt{T} + |u|c + |u|^2\gamma\right)$$

$\square$

## I   ROBUST LEARNING WITH UNKNOWN $G$

In this section, we present the regret bound to Algorithm 2 in Theorem 5. We assume all quantities are from Algorithm 2.

The proof in this section refers to a regret decomposition by substituting $\phi_t(w) = r_t(w) + a_t\psi(w)$ to Equation (10), where $\psi(w) = w^2$. This will allow us to identify four components that needed to be bounded, $R_T^{\mathcal{A}}(u)$, MAINTAIN, OFFSET$_1$ and OFFSET$_2$, in order to bound the true regret $R_T(u)$.

$$R_T(u) \leq \sum_{t=1}^{T} \langle \tilde{g}_t^c, w_t - u \rangle + a_t\psi(w_t) - a_t\psi(u) + r_t(w_t) - r_t(u)$$

$$+ \sum_{t=1}^{T} -a_t\psi(w_t) + a_t\psi(u) - r_t(w_t) + r_t(u) + \sum_{t\in\mathcal{P}} |g_t - \tilde{g}_t||w_t - u| + \sum_{t\notin\mathcal{P}} |g_t - \tilde{g}_t^c||w_t - u|$$

$$\leq \underbrace{\sum_{t=1}^{T} \langle \tilde{g}_t^c + \nabla r_t(w_t), w_t - u \rangle + a_t\psi(w_t) - a_t\psi(u)}_{R_T^{\mathcal{A}}(u)}$$

$$+ \underbrace{\psi(u)\sum_{t=1}^{T} a_t + \sum_t r_t(u) + |u|\sum_{t\in\mathcal{P}} |g_t - g_t^c| + |u|\sum_{t\notin\mathcal{P}} |g_t - g_t^c|}_{\text{MAINTAIN}}$$

$$+ \underbrace{\sum_{t\notin\mathcal{P}} |g_t - \tilde{g}_t^c||w_t| - \sum_{t=1}^{T} \alpha_t\psi(w_t)}_{\text{OFFSET}_1:\text{ due to adaptive clipping}} + \underbrace{\sum_{t\in\mathcal{P}} |g_t - \tilde{g}_t||w_t| - \sum_{t=1}^{T} \beta_t\psi(w_t) - \sum_{t=1}^{T} r_t(w_t)}_{\text{OFFSET}_2:\text{ due to corruption}}$$

$$(24)$$

**Theorem 5.** *Suppose $g_t, \tilde{g}_t$ satisfies assumptions in Equation (3) and (4). Algorithm 2 in response to $\tilde{g}_t$ with parameters: $\alpha = \epsilon/c, \gamma_\alpha = \gamma_\beta = \frac{\gamma}{2}$, for some $\epsilon, c, \gamma, \tau_G, \tau_D > 0$. Then Algorithm 2 guarantees a regret bound $R_T(u)$:*

$$R_T(u) \leq \tilde{O}\left[\epsilon + |u|c + |u|\max(\tau_G, G)\sqrt{T} + |u|^2\gamma\right] + \frac{4k^2G^2}{\gamma}\ln\frac{8k^2G^2}{c\gamma\tau_D} + c\tau_D + kG\tau_D$$

$$+ \frac{4(k+1)^2(G+h_T)^2}{\gamma}\left(1 + \ln\frac{h_{T+1}}{\tau_G}\right)\max\left(\left\lceil\log_2\frac{8G}{\tau_G}\right\rceil, 1\right)$$

Before providing the proof, we note a particular Corollary that yields "constant regret at the origin":

**Corollary 24.** *With $c = 2/\tau_D, \gamma = (k+1)^2$ and rest of parameters same as Theorem 5, Algorithm 2 guarantees a regret bound $R_T(u)$:*

$$\tilde{O}\left[\epsilon + \frac{|u|}{\tau_D} + kG\tau_D + |u|\max(\tau_G, G)\sqrt{T} + |u|^2(k+1)^2 + G^2\right]$$

Now, we proceed with the proof of Theorem 5.

*Proof.* The proof is by bounding each component in Equation (24).

OFFSET$_1$: due to adaptive clipping:

$$\text{OFFSET}_1 := \sum_{t\notin\mathcal{P}} |g_t - \tilde{g}_t^c||w_t| - \alpha_t|w_t|^2 \leq \sum_{t\notin\mathcal{P}} (G + h_t)|w_t| - \alpha_t|w_t|^2 \qquad (25)$$

For each fixed $t \in \bar{\mathcal{P}}$, we have $A_t|w_t| - \alpha_t|w_t|^2 \leq \sup_{X\geq 0} A_t X - \alpha_t X^2 \leq \frac{A_t^2}{4\alpha_t}$, where $A_t = G + h_t > 0$. Hence an upper bound to Equation (25) can be derived by substitute $\alpha_t$:

$$\text{OFFSET}_1 \leq \sum_{t\notin\mathcal{P}} \frac{(G+h_t)^2}{4\alpha_t}$$

$$
= \frac{1}{4\gamma_\alpha} \sum_{t \notin \mathcal{P}} \frac{(G+h_t)^2 h_{t+1}}{h_{t+1}-h_t} \left(1 + \sum_{i=1}^{t} \frac{h_{i+1}-h_i}{h_{i+1}}\right)
$$

$$
\leq \frac{(G+h_T)^2}{\gamma_\alpha} \left(1 + \sum_{i=1}^{T} \frac{h_{i+1}-h_i}{h_{i+1}}\right) \sum_{t \notin \mathcal{P}} \frac{h_{t+1}}{h_{t+1}-h_t}
$$

$$
\leq \frac{(G+h_T)^2}{\gamma_\alpha} \left(1 + \ln\frac{h_{T+1}}{\tau}\right) \sum_{t \notin \mathcal{P}} \frac{h_{t+1}}{h_{t+1}-h_t}
$$

$$
\leq \frac{(G+h_T)^2}{\gamma_\alpha} \left(1 + \ln\frac{h_{T+1}}{\tau}\right) |\bar{\mathcal{P}}| 2(k+1)
$$

$$
\leq \frac{(G+h_T)^2}{\gamma} \left(1 + \ln\frac{h_{T+1}}{\tau}\right) 4(k+1)^2 \max\left(\left\lceil \log_2 \frac{8G}{\tau_G}\right\rceil, 1\right)
$$

where the third line is due to $h_t$ being positive and non-decreasing by Lemma 18 (3). For the second to last line, a uniform bound on $h_{t+1}/(h_{t+1}-h_t) \leq 2(k+1), \forall t \notin \mathcal{P}$ was applied by Lemma 18 (5). Finally, an upperbound to $|\bar{\mathcal{P}}|$ by Lemma 18 (4) and the substitution of $\gamma_\alpha = \gamma/2$ was applied.

OFFSET$_2$: due to corruption:

The upper bound is obtained through two steps. In each step we aim to show:

$$
\text{OFFSET}_2 := \underbrace{\sum_{t \in \mathcal{P}} |g_t - \tilde{g}_t||w_t| - \sum_{t=1}^{T} \beta_t \psi(w_t)}_{\text{step 1: } \leq O(G^2 k \log(\max_t |w_t|))} - \sum_{t=1}^{T} r_t(w_t) \leq O(G^2 k \ln(\max_t |w_t|)) - \underbrace{\sum_{t=1}^{T} r_t(w_t)}_{\text{step 2: } \leq O(G^2 k)}
$$

By Lemma 19 property $(2)(3)$, we have

$$
\beta_t = \begin{cases} \gamma_\beta \cdot \dfrac{1/2}{1+\sum_{i=1}^{t} \frac{z_{i+1}-z_i}{z_{i+1}}}, & t = t_n, n \in [N] \\ 0, & \text{otherwise} \end{cases}
$$

Proceed with analysis to step 1, where second line is by Lemma 19 property (1) and value of $\beta_t$ displayed above:

$$
\text{step 1} := \sum_{t \in \mathcal{P}} |g_t - \tilde{g}_t||w_t| - \sum_{t=1}^{T} \beta_t \psi(w_t)
$$

$$
= \sum_{n=0}^{N} \sum_{t \in \mathcal{P} \cap \mathcal{T}_n} |g_t - \tilde{g}_t||w_t| - \sum_{n=1}^{N} \beta_{t_n}|w_{t_n}|^2
$$

$$
\leq \sum_{t \in \mathcal{P} \cap \mathcal{T}_0} |g_t - \tilde{g}_t||w_t| + \sum_{n=1}^{N} 2|w_{t_n}| \sum_{t \in \mathcal{P} \cap \mathcal{T}_n} |g_t - \tilde{g}_t| - \sum_{n=1}^{N} \beta_{t_n}|w_{t_n}|^2
$$

$$
\leq \tau_D \sum_{t \in \mathcal{P} \cap \mathcal{T}_0} |g_t - \tilde{g}_t| + \sum_{n=1}^{N} 2|w_{t_n}| \sum_{t \in \mathcal{P} \cap \mathcal{T}_n} |g_t - \tilde{g}_t| - \sum_{n=1}^{N} \beta_{t_n}|w_{t_n}|^2
$$

$$
\leq \tau_D \sum_{t=1}^{T} |g_t - \tilde{g}_t| + \sum_{n=1}^{N} 2|w_{t_n}| \sum_{t \in \mathcal{P} \cap \mathcal{T}_n} |g_t - \tilde{g}_t| - \sum_{n=1}^{N} \beta_{t_n}|w_{t_n}|^2
$$

$$
\leq kG\tau_D + \sum_{n=1}^{N} 2|w_{t_n}| \sum_{t \in \mathcal{P} \cap \mathcal{T}_n} |g_t - \tilde{g}_t| - \sum_{n=1}^{N} \beta_{t_n}|w_{t_n}|^2 \tag{26}
$$

where the third line is due to Lemma 19 property $(4)$, the forth line is due to Lemma 19 property $(2)$. Now we analyze each summands over $n$ in Equation (26). Considering a fixed $n \in [N]$:

$$
2|w_{t_n}| \sum_{t \in \mathcal{P} \cap \mathcal{T}_n} |g_t - \tilde{g}_t| - \beta_{t_n}|w_{t_n}|^2 \leq \sup_{X \geq 0} X \sum_{t \in \mathcal{P} \cap \mathcal{T}_n} 2|g_t - \tilde{g}_t| - \beta_{t_n}X^2
$$

$$= \frac{\left(\sum_{t \in \mathcal{P} \cap \mathcal{T}_n} |g_t - \tilde{g}_t|\right)^2}{\beta_{t_n}}$$

$$= \frac{2}{\gamma_\beta} \left(\sum_{t \in \mathcal{P} \cap \mathcal{T}_n} |g_t - \tilde{g}_t|\right)^2 \left(1 + \sum_{i=1}^{t} (z_{i+1} - z_i)/z_{i+1}\right)$$

$$\leq \frac{2}{\gamma_\beta} \left(\sum_{t \in \mathcal{P} \cap \mathcal{T}_n} |g_t - \tilde{g}_t|\right)^2 \left(1 + \sum_{i=1}^{T} (z_{i+1} - z_i)/z_{i+1}\right)$$

$$\leq \frac{2}{\gamma_\beta} \left(\sum_{t \in \mathcal{P} \cap \mathcal{T}_n} |g_t - \tilde{g}_t|\right)^2 \ln\left(1 + \frac{z_{T+1}}{z_1}\right)$$

Substitute to equation (26)

$$\text{step } 1 \leq kG\tau_D + \frac{2}{\gamma_\beta} \ln\left(1 + \frac{z_{T+1}}{z_1}\right) \sum_{n=1}^{N} \left(\sum_{t \in \mathcal{P} \cap \mathcal{T}_n} |g_t - \tilde{g}_t|\right)^2$$

$$\leq kG\tau_D + \frac{2}{\gamma_\beta} \ln\left(1 + \frac{z_{T+1}}{z_1}\right) \left(\sum_{n=1}^{N} \sum_{t \in \mathcal{P} \cap \mathcal{T}_n} |g_t - \tilde{g}_t|\right)^2$$

$$\leq kG\tau_D + \frac{2}{\gamma_\beta} \ln\left(1 + \frac{z_{T+1}}{z_1}\right) \left(\sum_{t \in \mathcal{P}} |g_t - \tilde{g}_t|\right)^2$$

$$\leq kG\tau_D + \frac{2}{\gamma_\beta} \ln\left(1 + \frac{z_{T+1}}{z_1}\right) (kG)^2$$

where the last step is due to $\mathcal{P} \subset [T]$ and the corruption model in Equation (4). By substituting $\gamma_\beta = \gamma/2$, $z_1 = \tau_D$, $z_{T+1} \leq \max(\tau_D, 2\max_t |w_t|)$, we obtained an upper bound to step 1:

$$\text{step } 1 := \sum_{t \in \mathcal{P}} |g_t - \tilde{g}_t||w_t| - \sum_{t=1}^{T} \beta_t \psi(w_t) \leq kG\tau_D + \frac{4k^2 G^2 \ln\left(1 + \max\left(1, \frac{2\max_t |w_t|}{\tau_D}\right)\right)}{\gamma}$$

$$\leq kG\tau_D + \frac{4k^2 G^2 \ln\left(2 + \frac{2\max_t |w_t|}{\tau_D}\right)}{\gamma}$$

Thus, an upper bound to $\text{OFFSET}_2$ is though obtaining an upper bound to step 2 defined as follows:

$$\text{step } 2 := \frac{4k^2 G^2 \ln\left(2 + \frac{2\max_t |w_t|}{\tau_D}\right)}{\gamma} - \sum_{t=1}^{T} \phi_t(w_t)$$

evoke Lemma 8 with $\alpha = \epsilon/c$

$$\leq \frac{4k^2 G^2 \ln\left(2 + \frac{2\max_t |w_t|}{\tau_D}\right)}{\epsilon} - c\max_t |w_t| + \epsilon$$

$$\leq \sup_{X > -2} \frac{4k^2 G^2}{\gamma} \ln(2 + X) - \frac{c\tau_D}{2} X + \epsilon$$

for $A, B > 0$, $A\ln(2 + X) - BX$ obtains its supremum at $X = A/B - 2 > -2$. Hence $\sup_{X > -2} A\ln(2 + X) - BX = A\ln(A/B) - A + 2B$. By substituting $A = \frac{4k^2 G^2}{\gamma}$, $B = \frac{c\tau_D}{2}$ we have

$$= \frac{4k^2 G^2}{\gamma} \ln \frac{8k^2 G^2}{c\gamma\tau_D} - \frac{4k^2 G^2}{\gamma} + c\tau_D + \epsilon$$

Thus step 1 and step 2 implies

$$\text{OFFSET}_2 \leq \epsilon + \frac{4k^2 G^2}{\gamma} \ln \frac{8k^2 G^2}{c\gamma\tau_D} - \frac{4k^2 G^2}{\gamma} + c\tau_D + kG\tau_D$$

MAINTAIN: comparator related term

This is first through Lemma 18 property (4) on $|\bar{\mathcal{P}}|$ is small

$$
\text{MAINTAIN} := \psi(u) \sum_{t=1}^{T} a_t + \sum_t r_t(u) + |u| \sum_{t \in \mathcal{P}} |g_t - g_t^c| + |u| \sum_{t \notin \mathcal{P}} |g_t - g_t^c|
$$

$$
\leq \psi(u) \sum_{t=1}^{T} a_t + \sum_t r_t(u) + |u| G k + |u|(G + h_T)(k+1) \max \left( \left\lceil \log_2 \frac{8G}{\tau_G} \right\rceil, 1 \right) \tag{27}
$$

It remains to show the first two terms in Equation (27) can be bounded by desired orders. For the first summand, $\sum_t a_t = \sum_t \alpha_t + \sum_t \beta_t$. Thus by Lemma 22

$$
\sum_t a_t \leq \gamma_\alpha \left( \ln \left( \ln \left( 1 + \frac{h_{T+1}}{h_1} \right) \right) + \gamma_\beta \ln \left( \ln \left( 1 + \frac{z_{T+1}}{z_1} \right) \right) \right)
$$

by Lemma 18 (3): $h_1 = \tau_G, h_{T+1} \leq \max(\tau_G, 8G)$, similarly by Lemma 19 (4): $z_1 = \tau_D, z_{T+1} \leq \max(\tau_D, 2 \max_t |w_t|)$

$$
\leq \gamma_\alpha \left( \ln \left( \ln \left( 1 + \max(1, \frac{8G}{\tau_G}) \right) \right) + \gamma_\beta \ln \left( \ln \left( 1 + \max(1, \frac{2 \max_t |w_t|}{\tau_D}) \right) \right) \right) = \tilde{O}(\gamma)
$$

where the last step is by substituting of $\gamma_\alpha = \gamma_\beta = \gamma/2$, and the fact that $\max_t |w_t| \leq \frac{\epsilon}{2G} 2^T$ guaranteed by Theorem 23.

The second term in Equation (27) can be upper bounded by Lemma 8 by substituting $\alpha = \epsilon/c$:

$$
\sum_{t=1}^{T} r_t(u) \leq 3c \ln T |u| \left[ \ln \left( 1 + \left( \frac{|u|}{\alpha} \right)^{\ln T} \right) + 2 \right] = \tilde{O}\left( c|u| \right)
$$

Thus,

$$
\text{MAINTAIN} \leq \tilde{O}\left( \gamma + c|u| + |u|(k+1) \max(\tau_G, G) \right)
$$

Combine results from Theorem 23 for $\tilde{R}_T^1(u)$, we complete the proof. $\qquad\square$

We also provide an dimension-free analogue to Theorem 5.

**Theorem 25.** *Suppose* $\mathbf{g}_t, \tilde{\mathbf{g}}_t$ *satisfies assumptions in Equation (3) and (4). Algorithm 4 has access to* $\tilde{\mathbf{g}}_t^c, h_{t+1}$ *in receiving* $\tilde{\mathbf{g}}_t$ *as provided by* FILTER. *By setting* $\mathcal{A}_{\mathbb{R}}$ *as Algorithm 2 with all parameters the same as that of Theorem 5. Then Algorithm 4 gaurantee the same regret as Theorem 5 with respect to* $\|u\|$.

*Proof.* By Theorem 10

$$
R_T(\mathbf{u}) \leq \sum_{t=1}^{T} \langle z_t, x_t - \|\mathbf{u}\| \rangle + \sum_{t=1}^{T} \|\mathbf{g}_t - \tilde{\mathbf{g}}_t^c\| (|x_t| + \|\mathbf{u}\|) + \frac{3\|\mathbf{u}\|}{2} h_T \sqrt{T}
$$

$$
= \sum_{t=1}^{T} \langle z_t, x_t - \|\mathbf{u}\| \rangle + \sum_{t \in \mathcal{P}} \|\mathbf{g}_t - \tilde{\mathbf{g}}_t^c\| (|x_t| + \|\mathbf{u}\|) + \sum_{t \in \bar{\mathcal{P}}} \|\mathbf{g}_t - \tilde{\mathbf{g}}_t^c\| (|x_t| + \|\mathbf{u}\|) + \frac{3\|\mathbf{u}\|}{2} G \sqrt{T}
$$

Since $|z_t| < h_t$ is guaranteed by Theorem 10, thus Theorem 5 can be used to bound the first three terms and we complete the proof. $\qquad\square$

## J  FENCHEL CONJUGATE

Here we collects basic properties of *Fenchel conjugate*, see reference such as Bertsekas (2009); Orabona (2019), and previously established Lemma used in Appendix E for completeness.

**Definition 26.** *Let $f : \mathbb{R}^d \to [-\infty, \infty]$, the Fenchel conjugate $f^*$ is defined as*

$$f^*(\theta) = \sup_{x \in \mathbb{R}^d} \langle \theta, x \rangle - f(x)$$

*the double conjugate $f^{**}$ is defined as*

$$f^{**}(\theta) = \sup_{x \in \mathbb{R}^d} \langle \theta, x \rangle - f^*(x)$$

**Theorem 27.** *Let $f : \mathbb{R}^d \to (-\infty, \infty]$*

  1.  *$f(x) = f^{**}(x), \forall x \in \mathbb{R}^d$ iff $f$ is convex and lower semicontinuous*

  2.  *$\langle \theta, x \rangle - f(x)$ achieves its supremum in $x$ at $x = x^*$ iff $x^* \in \nabla f^*(\theta)$*

**Lemma 28.** *(Theorem A.32 of Orabona (2019)) The Lambert function $W : \mathbb{R}^+ \to \mathbb{R}^+$ is defined as*

$$x = W(x) \exp(W(x)), \quad \text{for } x > 0$$

*and $W(x) > 0.6 \ln(1 + x)$ for $x > 0$.*

**Lemma 29.** *(Theorem A.3 of Orabona (2019)) Let $a, b > 0$, $f(x) = b \exp(x^2/2a)$. Then the Fenchel conjugate is*

$$f^*(\theta) = \sqrt{a}|\theta| \left( \sqrt{W\left(\frac{a\theta^2}{b^2}\right)} - \frac{1}{\sqrt{W\left(\frac{a\theta^2}{b^2}\right)}} \right)$$

*where $W(\cdot)$ is the Lambert function.*

**Lemma 30.**

$$\sqrt{x} - \frac{1}{\sqrt{x}} \geq \sqrt{\frac{x}{9}}, \quad \forall x \geq \frac{3}{2}$$

*Proof.* The proof is based on rearrange $x \geq \frac{3}{2}$, the condition is equivalent to

$$\left(1 - \frac{1}{3}\right) x \geq 1$$

Given $x > 0$, divide both side by $\sqrt{x}$

$$\left(1 - \frac{1}{3}\right) \sqrt{x} \geq \frac{1}{\sqrt{x}}$$

Rearrange and we complete the proof. $\square$

