# OpenReview forum: "Unconstrained Robust Online Convex Optimization"
_ICLR.cc/2025/Conference — Submitted to ICLR 2025_

### Official Review · Reviewer_BiJi · 2024-10-21

**Soundness:** 3
**Presentation:** 3
**Contribution:** 3
**Rating:** 6
**Confidence:** 3

**Summary:**

This paper considers the problem of online linear optimization with possibly corrupted gradients in an unbounded domain. For this problem, the authors further investigated two setups: known gradient upper bound $G$ and unknown $G$. For the first case, the authors obtained an $O(\|u\| G (\sqrt{T} + k))$, where $\|u\|$ denotes the norm of the unknown comparator that appeared in the definition of regret, and $k$ serves as an upper bound for the number of corruptions or the magnitude of corruptions. Note that the number $k$ is known prior to the algorithm. The authors have also provided a corresponding lower bound to prove the optimality of the obtained guarantee. For the unknown gradient norm upper bound, the authors obtained the above regret with an additive $(\|u\|^2 + G^2) k$ overhead. Finally, the authors have also applied their results to the applications of stochastic convex optimization with corruptions and DRO to validate the effectiveness.

**Strengths:**

The paper considers the meaningful problem of online learning with corruption. Advances in this problem can enhance the robustness of online algorithms. The presentation is clear and easy-to-follow. The example in Figure 1 is intuitive and well-motivating. The proof sketch in Section 4 is clear, intuitive, and easy to understand for readers without much background knowledge in this field. The obtained result for the known gradient norm upper bound case is optimal.

**Weaknesses:**

I am not an expert in online learning in the unbounded domain. As a result, I did not check the correctness of the proofs in this paper. The proof sketch in Section 4 is clear, as I have stated in the 'Strengths' part, and thus easy to follow. However, the analysis becomes much more complicated in the unknown gradient norm upper bound case. The current statements from Pages 8 to 10 are quite complicated and confusing. Actually, this part looks more like a draft or some proof that should be deferred to the appendix.  The current presentation is quite confusing for me. I suggest that the authors could revise this part to emphasize more about the exact difference in the analysis from that in the first case, but not state the proof once again.

**Questions:**

1. The paper [1] also considered some kind of corruption. Specifically, they studied the case where the strong convexity of online functions can be corrupted or degenerated to the convex case, and they used universal online learning algorithms to solve this problem. I think [1] is related to this paper in terms of corruption and can be introduced in the related work section.

2. I was wondering how strong the assumption of knowing $k$ is. Actually, since $k$ serves as an upper bound for both the corruption number and the corruption magnitude. Knowing it seems to be a pretty strong assumption. Can the authors provide further explanations on this issue? I also suggest that the authors could add more explanations about it in the revised version.

3. Can eq.(5) be simply rewritten as $\tilde{g}_t^c = \min\\{h_t, \|\tilde{g}_t|\\}$? The current version seems pretty complicated but not very necessary.


References:

[1] Contaminated Online Convex Optimization

---

> ### Author Response · Authors · 2024-11-19
>
> We appreciate feedback from Reviewer BiJi. Please refers to the global response where we made clarification on both Section 4 and Section 5 indeed rooted from a common motivation, but the analysis technique are very different with the second one being significantly more challenging. We have updated manuscript to improve readabilities, and updated Eqn (5) into a more natural representation.
>
> To address your remaining questions on Absent knowledge of $k$:
> - For the case of known $G$, Theorem 1 automatically tolerant problems with $k \le O(\sqrt{T})$, we can just set $k = O(\sqrt{T})$ (which is a huge amount of corruption) and will still retain sublinear regret. For the unknown $G$ case, Corollary 6 and 24 allows $k$ to be $O(\sqrt{T})$ and $O(T^{1/4})$, respectively while retaining sublinear dependence on $T$.
>
> - In general, dealing with large $k$ values is an interesting future direction - it’s not even clear if this is possible as all relevant works has assumed the knowledge of $k$ either in the form of Eqn (1) or Eqn (2).
>
> We hope these addresses your questions.

---

### Official Review · Reviewer_6WrH · 2024-11-03

**Soundness:** 3
**Presentation:** 1
**Contribution:** 2
**Rating:** 5
**Confidence:** 3

**Summary:**

This paper studies algorithms with good regret guarantees in a modification of the classical (unconstrained) online convex optimization setting where (roughly, since the definition is more nuanced) $k$ of gradients can be arbitrarily corrupted.  Previous work either studied stochastic corruptions and/or a setting where the feasible set of the algorithm/player is bounded. When the algorithm knows a bound $G$ on the norm of the gradients, this paper shows an algorithm whose regret guarantee against a comparator $u$ is (roughly) only a additive factor of $\tilde{O}(Gk \lVert u \rVert)$ worse compared to the regret guarantees of optimal algorithms in the uncorrupted setting. Furthermore, this paper also shows an algorithm that has similar regret guarantees even without prior knowledge of $G$, with an extra penalty of $\tilde{O}((\lVert u\rVert^2 + G^2)k)$ in the regret. Finally, the paper also provides matching lower bounds for the case of known $G$.

**Strengths:**

- The paper finds a natural problem with a clear technical difficulty: handling adversarially corrupted gradients in *unconstrained* problems in online convex optimization. It was surprising that this problem was not previously studied outside of the stochastic case;
- The definition of $k$ is simple yet nicely covers two slightly different notions of robustness in OCO simultaneously;
- The final regret bounds are asymptotically optimal, and the matching lower bounds show that we cannot improve the bounds by much;

**Weaknesses:**

- I believe one of the main weaknesses of this paper is presentation. The first algorithm (for known $G$) heavily builds on the work of Zhang and Cutkosky (2022). This is not a problem by itself, but the paper does not do a great job os describing the main modifications needed , and I found myself relying on reading the other paper to understand the main ideas of the algorithm, which is not ideal. For the case of unknown $G$ the authors seem to combine (and improve on) different ideas from previous work. However, the presentation is convoluted, and it is hard to understand both the algorithm and the main technical contributions (I will expand on this in the questions);

- The examples are not great: the first one is not interesting (at least from what I could understand), since the reduction of OCO to stochastic optimization is almost just matter of taking expectations of the OCO guarantees. And for the second example I could not understand whether the results the papers gets are interesting or not since there is no discussion on what kinds of results exist on the literature of the area already;

- At some points in the main text there is a confusion between working in 1D and general dimension, which makes the description of the algorithms and results very confusing;


---
**Summary**: The problem the paper studies is interesting and, in some sense, natural. Also, the regret guarantees the paper presents depend mildly on the corruption level $k$, and the paper also shows lower bounds that their regret bounds are asymptotically optimal.  However, the techniques on the first algorithm build heavily on previous work, and the paper does not do a great job on describing the main ideas and differences with previous work. For the second algorithm, the presentation is quite convoluted and it is hard to verify some of the claims in the paper. The presentation is also confusing at a couple important points, making it even harder to understand the paper. Finally, the examples presented do not add much to the paper.

Currently, I believe the contributions of the paper are valuable, even if heavily based on previous work. Yet, the presentation could be greatly improved, which is even more important given the similarity with the techniques of previous work. Thus, currently I am not comfortable vouching to accept the paper. I am sending a few questions that hopefully will help me better understand the contributions of the paper, but I'd still strongly recommend improving the presentation of the paper;

**Questions:**

I believe if the authors could help clarify a few of the key ideas and contributions in the algorithm, it could help me and the other reviewers better appreciate your work.


- **The case of known $G$**: The algorithm in this case is a modification of ideas proposed by Zhang and Cutkosky (2022). However, it is not clear from the paper what are the key differences between the analysis in the stochastic case done by Zhang and Cutkosky and the one done in this paper.

I know the space constraints play into this, but reading pieces of Zhang and Cutkosky (2022) clarified many pieces of the algorithm for me that were not clear from the algorithm. A small example: on line 6 of Algorithm 1 we need to find a point $w_t$ that satisfies an equation, but the paper never discusses computational complexity or, more importantly, why a solution to this is guaranteed to exist. This is luckily done by Zhang and Cutkosky (2022), but it is not even discussed in the paper. My question for the authors here is: could you briefly explain what are the key modifications to the algorithm and/or analysis from the stochastic case from Zhang and Cutkosky to this adversarial corruption setting? If the algorithm is barely different from their algorithm for sub-exponential noise, it would be even more interesting and should even be clarified in the paper, because then a contribution of the paper would be showing that the same algorithm also works for this seemingly harder case!

- **The case of unknown $G$**: This case seems to be about estimating $G$ on-the-fly, doing so by using the algorithm for known $G$ with a couple of ideas from previous work (an improvement on the filtering strategy from van Erven et al., 2021, and the Epigraph Based regularization from Cutkosky and Mhammedi, 2024). This was a section that was quite hard to understand, and I have a few questions in the hope of better understanding the contributions of this section.

1) On the paragraph starting on line 399, you mention you improve on the original Filter strategy. Besides the memory usage, are there other improvements? Are these improvements crucial for the algorithm? (i.e., a black-box use of Filter could lead to similar bounds?)

2) When using the Epigraph-based regularization the paper also uses a reduction from e Cutkosky & Orabona (2018) to maintain the constraints, but it seems that this reduction is explicitly written into the algorithm. Is the use of this reduction the same described by  Cutkosky & Mhammedi (2024), section 3.3? If not, what are the main differences? Also, since this is a reduction, couldn't algorithm 2 be described in two steps (such as the different protocols in Cutkosky & Mhammedi, 2024)? I think this would simplify the presentation greatly;

- **Confusion in the description of the algorithm**: On eq. 5, $\tilde{g}_t^c$ might be a scalar or a vector, depending on the case. I think you meant to present this only for the 1D case. This is a key definition for the algorithm, so it is important for it to be correctly described.

- **On the second example**: Are there comparisons with results in the area of DRO that can help us understand the relevance of the application of this results to this problem?

---

> ### Author Response · Authors · 2024-11-19
>
> We appreciate the detailed feedback from Reviewer 6WrH. For novelty concerns, and presentation issues please see global response and updated manuscript.
>
> To address your remaining questions:
>
> ## Unknown G case necessity of new design of ``filter`` in replacing van Erven (2021):
> - Our ``filter`` draws inspiration from van Erven (2021) with, but saves space complexity and removes the need to sort, insert and remove elements.
> - More importantly, the new design of ``filter`` is critically necessary for analysis, and the simpler van Erven (2021) would not suffice. A key distinction is that our ``filter`` causes the  “maximum” value recorded by streaming incoming $\tilde g_t$ to change very infrequently. This property is essential for allowing our regularization weights $\alpha_t$ to simultaneously achieve
> $$\sum_t \alpha_t = \tilde{O}(1), \sum_{t: \alpha_t > 0} \frac{1}{\alpha} \le  \tilde{O}(\log(h_T / h_1))$$
> which is critical in achieving regret in Theorem 5.
> - As a result, a direct application of filtering from van Erven (2021) cannot achieve the regret bound established in Theorem 5. But our ``filter`` can be used in the set-up in van Erven (2021) by simply clipping gradients to 0 whenever our clipping would be evoked. This will yield the same regret bound in van Erven (2021) for their constrained setting on the subset regret they defined.
>
> ## DRO:
> - DRO itself is a broad research area not limited to empirical risk minimization (ERM) problems. Additionally, there exists a diverse range of uncertainty sets $\mathcal{P}_k$ (for example see [1] section 2). Each notion of uncertainty sets $\mathcal{P}_k$ receives dedicated attention within the DRO community.
> - Our results naturally apply in ERM setting in DRO when uncertainty set $\mathcal{P}_k$ is defined as Total variation ([1] section 2.2.3) and KL divergence ([1] section 2.2.1)
> - To the best of our knowledge, the most directly comparable result is [2] who also considered Total variation as part of their assumptions in the known $G$ and constrained regime. Their results exhibit an additional additive factor of $k$ (in their notation as $\rho$). Our results from the known $G$ case matches their bound and in addition holds for the unconstrained setting.This is not surprising due to the lower bound in Theorem 2.
> - The remaining cited papers in relevant sections focus on alternative uncertainty sets or problems outside ERM, Many of them are empirical applications due to the growing interest in multimodal models. Given the demenstrated empirical success and the lack of theoretical investigations, our application example offers some theoretical insight for this set-up.
>
>
> We hope these clarifications addresses your questions.
>
> [1]: Sun, X.A., Conejo, A.J., Sun, X.A. and Conejo, A.J., 2021. Distributionally Robust Optimization. Robust Optimization in Electric Energy Systems, pp.131-204.
>
> [2]: Namkoong, H. and Duchi, J.C., 2016. Stochastic gradient methods for distributionally robust optimization with f-divergences. Advances in neural information processing systems, 29.

---

> > ### Comment · Reviewer_6WrH · 2024-11-19
> > **Follow-up questions and comments**
> >
> > I would like to start by thanking the authors for taking the time to answer to the questions that I and other reviewers posed. The changes to Section 5 together with the author's rebuttal really helps highlight the key factors that this paper brings to the table to make everything work! I still feel that the section is somewhat hard to follow, but the modifications do help a lot.
> >
> > However, my previous points about (1) the lack of clarity on the contributions in the case of known $G$ and (2) DRO application not being well-placed in the literature are still there.
> >
> > **(1) Lack of clarity on the contributions when $G$ is known (Sec. 4)**. The rebuttal made me a bit more confused about the differences of the techniques used in Zhang and Cutkosky (2022) to the ones proposed in Sec. 4. In the way that the paper is currently written, it sounds like the techniques are similar ("We take inspiration from..." is used a few times) but significant changes were necessary, so much so that the proofs were re-done and the algorithm described from scratch. However, in the rebuttal, the authors wrote
> >
> > > In the known G case (sec 4.1), the primary challenge lay in reformulating the problem using clipping and $\tilde{g}_t^c$ in such a way that it was clear that the techniques of Zhang and Cutkosky (2022) would be applicable.
> >
> > Thus, is it the case that the techniques used in Section 4 are *identical* to Zhang and Cutkosky (2022), except for rewriting/reformulation of the problem? In this case, it seems like Sec. 4 should describe a reduction/generalization. I think this lack of clarity in one of the main contributions of the paper is not ideal.
> >
> > **(2) DRO application is not well-placed in the literature**: Even with the explanation the authors provided about the DRO application, I could not place the results in the literature. The discussion in the rebuttal made me a bit confused nNot clear why ERM was mentioned, and the sections referenced in the book mentioned did not even mention Total variation or KL divergence, so I could not understand the relevance to the discussion), but there is one main problem in the discussion of the DRO application.
> > - *Main problem*: it seems that the setting these results are applied to is the online setting proposed by Namkoong and Duchi (2016), thus I would expect their results to be directly comparable to the results from this paper. I did not have the time to parse the paper properly, but skimming it quickly could not identify which results would be the comparison point to the results of Corollary 4. Based on the rebuttal, you are claiming that their results both have a worse dependency on $k$ AND do not apply to the unconstrained setting? So is Corollary 4 a strict improvement of results from Namkoong and Duchi (2016)? This is the key discussion that is lacking in this section.
> >
> > Since the other reviews did not mention the applications of the results, the authors should consider whether it is worth trying to keep this applications in the paper.

---

> > > ### Author Response · Authors · 2024-11-20
> > >
> > > **(1) Known $G$ contribution**:
> > >
> > > - In the case of known $G$, we use essentially the same algorithm applied to our specific setting. The high-level intuition of why it works is the same, but the actual analysis diverges due to the investigated context being different. Our results do not immediately follow any results from Zhang & Cutkosky (2022), so we cannot frame it as a reduction from Zhang & Cutkosky (2022). And our results do not generalize to their setting so we cannot phrase it as a generalization.
> > > - The proof is “re-done” only for the necessary components and is documented in Appendices B and C, Specifically:
> > > 1.  Appendix B builds on reduction from Cutkosky (2019b) but necessitates using a different base learner for compatibility with the unknown $G$ case (Algorithm 2). Consequently, Algorithm 1 became a special case of Theorem 7. The rationale and expectation has been explicitly outlined in Appendices A and B. It is standard and necessary to re-derive the reduction in cases where requiring different base learners, if this reduction with those learners has never appeared in literature.
> > > 2. For Appendix C, we stated Lemma 8 as the general form from Zhang & Cutkosky (2022), and only being specific on exact constants and relaxation used for our context.
> > >
> > > **(2) DRO well-placed issue**:
> > > - We apologize for the incorrect reference to [1], and the intended reference should have been https://www.arxiv.org/abs/2411.02549 which provides a comprehensive review of diverse research topics and formulations in DRO. This reference was meant to illustrate the broad scope of DRO research and help narrow down comparable topics.
> > > - We do not find exactly the same online setting result in literature, so this is the first result in online setting. Upon reflection with the reviewer’s detailed comments, we acknowledge that our previous claims regarding the relationship with Namkoong & Duchi (2016) may have been overstated. To clarify:
> > > - Namkoong & Duchi (2016) solves an Robust ERM problem in a constrained setting and uses online learning for algorithm design and as an analysis tool. And Robust ERM with $f$-divergence is the closest topic.
> > > - Namkoong & Duchi (2016) achieves an additive factor of $k$ in comparison to standard ERM problem without the distribution robustness. Our results also achieve an additive factor of $k$ in comparison to standard online learning problems without the context of distribution robustness. Our result is not directly comparable to theirs.
> > > - For further details: Namkoong & Duchi (2016) demonstrate their results in the constrained setting with bounds such as the following (Page 2, starting from the last nine lines): https://web.stanford.edu/~jduchi/projects/NamkoongDu16.pdf
> > > “In particular, we show (for many suitable divergences f) if $l_i$ is L-Lipschitz and $\chi$ has radius bounded by R, then our procedure requires at most $O(\frac{R^2L^2 + \rho}{\epsilon^2})$ iterations to achieve an $\epsilon$-accurate solution to problem (1), which is comparable to the number of iterations required by SGD [23].” The relevant notations are introduced at the bottom of Page 1 of their work.
> > > - If the reviewer believes the inclusion of this application diverts focus, we are open to omitting this discussion in the final revision to maintain the paper’s clarity and core contributions.

---

> > > > ### Comment · Reviewer_6WrH · 2024-11-26
> > > >
> > > > I appreciate the authors being patient and taking the time to address the points I raised. I am sorry for taking some time to reply again, but I have limited time to engage in these discussions.
> > > >
> > > > For the case of the contribution for the case of known $G$, I think the explanation given by the reviewers makes sense, but I will try to take the time to take a look at the proofs more carefully so I can build a more solid opinion. However, I might not be able to do so before the end of the discussion period.
> > > >
> > > > As for the DRO case, the explanation the reviewers gave is helpful. It is also worrisome that the original version of the paper didn't quite capture the correct connection with the work of Namkoong and Duchi. Nonetheless, it is positive that the review process helped clarify this part of the paper.
> > > >
> > > > Concerning the potential removal of the DRO discussion, I want to be clear: this component does not significantly influence my overall assessment of the paper. The authors should keep this result if they believe in its value, independent of my (or any other reviewers') opinions. The decision to keep or remove this section should be guided by the authors' scientific judgment rather than by reviewing dynamics. Hopefully the reviewers can help the authors see things from a new light, but reviews should only guide the authors on making this decisions.

---

### Official Review · Reviewer_eaW4 · 2024-11-04

**Soundness:** 4
**Presentation:** 4
**Contribution:** 3
**Rating:** 6
**Confidence:** 4

**Summary:**

The authors study online convex optimization under corrupted gradient feedback in an unconstrained domain. The proposed algorithm requires prior knowledge of "a measure of total corruption" $k$ and achieves a regret bound of $\mathcal{O}(|| u || G (\sqrt{T} + k))$ for any comparator $u$, assuming the gradient norm bound $G$ is known for the uncorrupted rounds. If $G$ is not known, they propose a filtering approach to guarantee a regret bound of $(|| u ||^2 + G^2) k$. They also provide matching lower bounds (up to logarithmic factors) for any choice of $u$.

**Strengths:**

The paper is well-written, effectively motivating the problem and clearly articulating the associated challenges. The techniques and results are presented in an understandable and clear manner. The work draws significant inspiration from two existing works: it leverages the composite loss function method from Zhang & Cutkosky (2022) to manage large corrupted gradients and employs filtering techniques from van Erven et al. (2021) to address unknown bounds on the gradient norms of uncorrupted rounds. Especially the proposed work is able to reduce the space complexity of filtering techniques of van Erven et al. (2021).

**Weaknesses:**

In my view, the reliance on established techniques impacts the overall novelty of the work. Specifically, the major technical tools employed are well-known, which somewhat limits the innovative contribution. It would significantly strengthen the paper if the authors could elaborate on the primary technical innovations in Section 4.1. In particular, it would be helpful to clarify how their approach in this section differs from or improves upon the methods presented by Zhang & Cutkosky (2022).

**Questions:**

Please see the previous section.

---

> ### Author Response · Authors · 2024-11-19
>
> We appreciate feedback from reviewer eaW4. In terms of technical or intellectual novelties, please see global response for a detailed explanation, and the updated manuscripts in better articulating the unknown $G$ case.
>
> We hope the response addresses your concerns.

---

> > ### Comment · Reviewer_eaW4 · 2024-11-26
> >
> > Thank you for your response. I have decided to maintain my score.

---

### Official Review · Reviewer_DXKr · 2024-11-07

**Soundness:** 3
**Presentation:** 3
**Contribution:** 3
**Rating:** 6
**Confidence:** 4

**Summary:**

This paper studies unconstrained OCO with adversarial corruptions. The auhors provide algorithms that ensures $O(||u||G\sqrt{T}+||u||Gk)$ regret bound, where $k$ is the number of corruptions. They also provide matching lower bounds.

**Strengths:**

The problem:

This paper is the first to study unconstrained OCO with adversarial corruptions, which is a new and novel problem.

The motivation is clear and makes sense to me, as in practice we indeed might only able to observe biased gradients.

The contribution:

The authors successfully provide an algorithm which is of order $O(||u||G\sqrt{T}+||u||Gk)$ for the case when G (the upper bound for gradient) is known, which matches the optimal rate for parameter free OCO when the number of corruptions $k=O(\sqrt{T})$. There is also a lightly worse bound for the case where $G$ is not known. Finally, the authors also a matching lower bound for this problem.

The presentation:

The paper is in general easy to follow, and the main ideas of the methods are well explained.

**Weaknesses:**

The problem:

The problems studied here is a combination of the well-studied unconstrained OCO and OCO with corruption. Note that OCO with stochastic corruption has been studied, and many key ideas of this paper are motivated by them.


The methods:

The main technique used here to deal with corrupted gradients is very similar to that in Zhang & Cutkosky (2022). To be more specific, similar to the stochastic setting, the regret of this problem can be decomposed into two terms: one "easy" term, e.g., the first term in (6), a standrd OLO problem with bounded gradient, and one "bias" term (e.g., the second term in (6), which related to the gap between the clipped gradient and the true gradient). For the bias term, by basic Cauchy-Swhurz inequality (eq. (7)), one can notice that it is dominated by max_t ||\w_t||, that is, the max norm of the decisions. The same term also appears in  Zhang & Cutkosky (2022) (page 4, the NOISE term). Following Zhang & Cutkosky (2022), the authors introduced the same surrogate loss.


The presentation:

2) In Sections 4 and 5, the authors sometimes use ||\cdot\|| to denote the norm, sometimes use |\cdot| (e.g., compare (5) and (7)). Please make it clear in the begining of Section 4 that the discussion is for W=R, but it can be extended to R^d.

3) Line 215 "(in fact, potentially exponential in t)": I am not sure about the meaning of this sentence. Note that this paragraph is about regret decomposition, and no algorithm is discussed. So why it can exponential in t? The same sentence appears in Zhang & Cutkosky (2022) (first sentence in Page 5), but in Zhang & Cutkosky (2022) it makes sense there because it is in a different context (discussing an algorithm).

**Questions:**

In Section 4.2, can the authors provide some more explanations on how the lower bound is proved?

---

> ### Author Response · Authors · 2024-11-19
>
> We appreciate reviewer's DXKr feedback, for novelty concerns, and presentation issues please see global response and updated manuscript.
>
> To address your remaining questions:
>
> ###  $|w_t|$ grows exponential in $t$:
>
> - Thank you for raising the confusion, this was originally shown by Zhang and Cutkosky (2022) Lemma 8. It is a result for general unconstrained online learning algorithms, independent of the context of either statistical or adversarial corruptions.
>
> ### Section 4.2, lower bound:
>
> - The proof was deferred to Appendix E, which only relies on Lemma 12. The core idea is randomization, where gradients are radamarker random variables. This is often used to prove online learning algorithms lower bounds.
>
> - In unconstrained online learning, comparators $u$ with large magnitude are oftenly regarded as difficult. One might think when $|u|$ is small, the corruption problem is trivial. Thus we have Theorem 2 as lower bound to show the additive factor of $k$ is unavoidable for comparators $u$ with $\textbf{any}$ magnitude.
>
> We hope these clarifications addresses your concerns.

---

### Author Response · Authors · 2024-11-19
**Global Response**

We appreciate reviewers' feedback and have updated the manuscript to incorporate critical changes labelled in red. We address commonly appeared questions here.

## [Novelty] Main difference from Zhang and Cutkosky (2022):

Several reviewers addressed the concern that the similarity between this paper and Zhang and Cutkosky (2022), whose techniques indeed play an important role in our results.

In the known G case (sec 4.1), the primary challenge lay in reformulating the problem using clipping and $\tilde g_t^c$ in such a way that it was clear that the techniques of Zhang and Cutkosky (2022) would be applicable.

In the unknown G case (sec 5.2), there were significantly greater technical hurdles to overcome. In this case, the problem is that not only is $G$ unknown at the start of the game, it is **never revealed even at the end of the game**. We do make use of recent regularization schemes inspired by Zhang and Cutkosky (2022) and Cutkosky and Mhammedi (2024),  but on their own would not have solved our problem because they *also* require knowledge of $G$, either up-front, at least after the current round.

To fix this, we add two new “maximum-tracking” schemes as well as adding a carefully-tuned quadratic regularizer $(\alpha_t + \beta_t)\|w_t\|^2$. As is usual with regularization, we want to add as little regularization as possible to achieve our goals. This is where our new maximum-tracking schemes come in - they enable us to identify rounds in which corruption has caused undue harm, and make corrections on-the-fly.

Specifically, ``filter`` tracks gradients provides $h_{t+1}$ to estimated $max_\{i \le t\} | g\_i |$ (although it only sees $\tilde{g}\_i$ not even $g_i$), and ``tracker`` keeps track of a slowly-changing estimate of $\max_t \|w\_t\|$ as $z_{t+1}$. As motivated in Eqn (10) until the end of page 8, ``filter`` controls ''truncation error'' and ``tracker`` controls ''corruption error''. The **fundamental difference** is that the round which incurs ''truncation error'' is completely predictable (denoted as $\bar{\mathcal{P}}$), but the latter is not since corruption can happen at every single round.

Those two mechanisms are critical and use different analysis techniques in order to achieve regret in Theorem 5. Roughly speaking, with reference to Eqn (14) we can show (details in Appendix I line 1611 - line 1727):

$$ \textbf{offset} \lesssim \tilde{O} ( \sum\_{t: \alpha\_t > 0 }  \frac{1}{\alpha\_t}  +  \sum\_{t: \beta\_t >0} \frac{1}{\beta\_t} ) \le \tilde{O}(G^2k)
$$

where $\alpha_t, \beta_t$ sequences are defined in Eqn (11) and (12). This is guaranteed by ``filter`` and  ``tracker`` such that rounds that $\alpha\_t, \beta\_t >0$ are small to counteract with predictable rounds that incurs ''truncation error'' and potentially every single rounds which accumulating ''corruption error''.

Algebraically, to achieve the above goal in $\textbf{offset}$, we require $\alpha\_t, \beta\_t$ sequences to have interesting properties. Using $\alpha_t$ as an example, we want to achieve the following seemingly-impossible pair of identities:

$$\sum_t \alpha_t = \tilde{O}(1), \sum_{t: \alpha_t \neq 0} \frac{1}{\alpha} \le \tilde{O}(\log(h_T / h_1))$$

The first inequality was achieved by Cutkosky and Mhammedi (2024), but the the second is the key in achieving the desired bound for $\textbf{offset}$ and is completely new. Achieving both bounds simultaneously is not trivial from both algorithm design and analysis point of view.


## Clarity of Presentation regarding to Unknown G case:
We appreciate the reviewers' feedback regarding the presentation of Section 5.
To address these concerns, we have revised the manuscript to clarify the connection of section 5.1 to the main results, and the key takeaways needed for the following section. For section 5.2, we have updated the ordering of some equations, which better connects section 5.1

## Presentation of Algorithm 2:
As reviewer 6WrH spotted, Algorithm 2 involves our new techniques along with several different reductions. Specifically, we employed Cutkosky & Orabona (2018), Cutkosky (2019b) and Cutkosky and Mhammedi (2024) due to specific technical challenges needed to overcome. That means it is not identical to Cutkosky and Mhammedi (2024) section 3.3 (their algorithm 2). We have updated the manuscript with highlights in linking algorithm lines with each reduction.

## Confusion in Switching between $\mathbf{R}$ and $\mathbf{R}^d$:
The reduction technique from Cutkosky & Orabona (2018) is standard in designing unconstrained online learning algorithms operating on $\mathcal{R}^d$, allowing us to focus on learners in $\mathcal{R}$. Although we stated in lines 189-191 that Sections 4.1 and 5 consider $\mathcal{W} = \mathbf{R}$ with results extending to $\mathcal{W} = \mathbf{R}^d$, we will make the entire main text as $\mathcal{W} = \mathbf{R}$ to avoid further confusion.

---

### Meta-Review · Area_Chair_KD5B · 2024-12-12

**Metareview:**

This paper addresses the problem of unconstrained online convex optimization (OCO) with adversarial corruptions, providing algorithms that offer regret bounds dependent on the number of corruptions. The paper presents results for two cases: when the gradient norm upper bound is known and when it is unknown, along with matching lower bounds. The findings are applied to stochastic convex optimization and distributionally robust optimization (DRO).

Reviewers appreciated the novel problem and valuable theoretical contributions of the paper but raised concerns about the heavy reliance on existing techniques, such as those from Zhang & Cutkosky 2022. The presentation was found to be unclear in sections dealing with the unknown gradient norm, making it difficult to distinguish the paper's novel contributions. Additionally, reviewers emphasized the need for clearer examples and a more thorough explanation of assumptions to improve the overall clarity and impact of the paper.

**Additional Comments On Reviewer Discussion:**

After discussions, the authors have addressed some of the concerns. However, the reviewers still believe that, although the paper makes a technical contribution by providing regret bounds for unconstrained OCO with corruption, the work heavily draws from existing research. In particular, the algorithm described in Section 4.1 closely resembles the one from Zhang & Cutkosky (2022), but the paper does not clearly explain the differences or innovations in their approach. Additionally, the paper's reliance on Epigraph-Based Regularization (EBR) from Cutkosky & Mhammedi (2024) is not well-explained, making it difficult to fully grasp its role. Overall, the paper requires significant revisions, particularly in terms of clarifying the novel contributions and improving the clarity of exposition.

---

### Decision · Program_Chairs · 2025-01-22

Reject